# TOPO-DIFFUSION: TOPOLOGICAL DIFFUSION MODEL FOR IMAGE AND POINT CLOUD GENERATION

## ABSTRACT

Diffusion models represent an emerging topic in generative models and have demonstrated remarkable results in generating data with high quality and diversity. However, diffusion models suffer from two limitations (1) the notorious problem of computational burden such as large sampling step and train step needed due to directly diffusion in Euclidean space and (2) a global-structure representation for each sample that is implicitly considered in the process due to their very limited access to topological information. To mitigate these limitations, recent studies have reported that topological descriptors, which encode shape information from datasets across different scales in a topological space, can significantly improve the performance and stability of deep learning (DL). In this study, inspired by the success of topological data analysis (TDA), we propose a novel denoising diffusion model, i.e., Topo-Diffusion, which improves classical diffusion models by diffusing data in topology domain and sampling from reconstructing topological features. Within the Topo-Diffusion framework, we investigate whether local topological properties and higher-order structural information, as captured via persistent homology, can serve as a reliable signal that provides complementary information for generating a high-quality sample. Theoretically, we analyze the stability properties of persistent homology allow to establish the stability of generated samples over diffusion time steps. We empirically evaluate the proposed Topo-Diffusion method on seven real-world and synthetic datasets, and our experimental results show that Topo-Diffusion outperforms classical diffusion models across all the evaluation metrics in fidelity and diversity of sampled synthetic data.

## 1 INTRODUCTION

Diffusion models have emerged as the new state-of-the-art family of deep generative models. Recently, there has been a growing interest in generating synthetic images and data by diffusion models, which have been proven to be useful in various discriminative tasks including image segmentation (Graikos et al., 2022; Rombach et al., 2022; Wolleb et al., 2022), classification (Zimmermann et al.), text-to-image synthesis (Gu et al., 2022), and anomaly detection (Wolleb et al., 2022). They have broken the long-time dominance of generative adversarial networks (GANs) in handling challenging tasks of image synthesis, computer vision, natural language processing, temporal data modeling, multi-modal modeling, robust machine learning, and interdisciplinary applications in fields such as computational chemistry and medical image reconstruction (Croitoru et al., 2023). For instance, Rombach et al. (2022) introduce latent diffusion models that scale more gracefully to higher dimensional data and further reduce computational costs. Diffusion model involves two interacted processes (1) forward process that transforms the data distribution into a simpler prior distribution, such as a Gaussian distribution (Ho et al., 2020; Baranchuk et al.; Nichol & Dhariwal, 2021; Dhariwal & Nichol, 2021); (2) corresponding reverser process, in which a generative neural network is utilized for recovering the original input data from the diffused (degraded) data, via variational probabilistic optimization, score-matching (Song & Ermon, 2019), stochastic differential equations modeling (Song et al., 2020), etc. Denoising diffusion probabilistic models (DDPMs) (Ho et al., 2020; Baranchuk et al.; Nichol & Dhariwal, 2021; Dhariwal & Nichol, 2021) are latent variable models that employ latent variables to estimate the probability distribution. Another group of diffusion models are based on training a shared neural network via score matching to estimate the gradient of the log density (i.e., score function) of the perturbed data distribution at different noise

levels (Song & Ermon, 2019). As an alternative strategy, the stochastic differential equations (SDEs) based diffusion model represents diffusion via continuous forward and reverse SDEs, and can be viewed as a generalization over DDPMs and score matching based diffusion model (Song et al., 2020). In this work, we consider a new aspect to model the diffusion process by structural degrading and reconstruction via topological features.

Hence, in the last few years, we observe an increasing interest in enhancing the performance of diffusion models by introducing topological-based diffusion models. Specifically, these topological-based diffusion models typically integrate deep generative models with persistent homology (PH) representations of the learned objects, typically in the form of topological layer or topological loss in deep generative models (Wang et al., 2020). PH is a key tool in topological data analysis (TDA) which retrieves evolution of the shape patterns in the observed data along various user-defined geometric dimensions (Hofer et al., 2019; Edelsbrunner et al., 2000; Zomorodian & Carlsson, 2005). By "shape" here, we broadly refer to the data properties which are invariant under continuous transformations such as bending, stretching, and twisting. The main goal of PH is to retrieve the loss underlying structural properties (Wasserman, 2018). That is, the representations derived from PH enable us to identify and study the hidden topological descriptors of the shape of complex structured and unstructured data.However, existing topological-based generative models have significant drawbacks that need to be improved. For example, to learn topological properties from images, the topology-aware GAN (Wang et al., 2020) cannot simultaneously capture local and global topological structures. Liu et al. (2019) have addressed the issue of preserving both local and global topological features, but their method suffers from high complexity and thus is limited in its applications.

To tackle the aforementioned drawbacks of existing topological-based diffusion models, we propose a novel Topological Denoising Diffusion model, namely Topo-Diffusion, to efficiently incorporate local topological information into a denoising diffusion model. Contrary to previous works, we consider topological degrading and reconstruction processes in data generation from diffusion models for identifying structural features from data. Theoretically, we prove the theoretical stability guarantees of the Topo-Diffusion model. Empirically, we conduct comprehensive evaluation of Topo-Diffusion and demonstrate that it outperforms several baseline models, using multiple quantitative metrics on synthetic and real-world image datasets.

The primary contributions of this work can be summarized as follows:

- Topo-Diffusion is the first approach bringing the concepts of persistent homology and topological representation learning to diffusion generative models. We derive theoretical stability guarantees of the proposed topological features of generated data.
- We demonstrate the utility of our proposed Topo-Diffusion in conjunction with generating samples from point clouds to real-world images. Our findings show that Topo-Diffusion delivers better performance across all the evaluation metrics on various datasets.

The rest of this manuscript is organized as follows. We review related work in Section 2 and introduce preliminaries in Section 3. We describe the details of Topo-Diffusion and present theoretical results in Section 4. We summarize our empirical experiments with evaluation results in Section 5. Finally, we conclude the manuscript and point to future directions in Section 6.

## 2 RELATED WORK

**Persistent homology.** Persistent homology (PH) (Edelsbrunner et al., 2000; Zomorodian & Carlsson, 2005) is a suite of tools within TDA that has shown a great promise in a broad range of domains including bioinformatics, material sciences and social networks (Otter et al., 2017; Carlsson, 2020; Dey & Wang, 2022). One of the key benefits of PH is that it can capture subtle patterns in the data shape dynamics at multiple resolution scales. PH has been successfully integrated as a fully trainable topological layer into various machine learning and deep learning models (Pun et al., 2018), addressing such tasks as image classification (Hofer et al., 2017), 2D/3D shape classification (Bonis et al., 2016; Hofer et al., 2019), molecules and biomolecular complexes representation learning (Cang et al., 2018), graph classification (Horn et al.; Chen & Gel, 2023), and spatio-temporal prediction (Zeng et al., 2021; Chen et al., 2021). For example, Carrière et al. (2020) build a neural network based on the DeepSet architecture (Zaheer et al., 2017) which can achieve an end-to-end

learning for topological features. Cang et al. (2018) introduce multi-component persistent homology, multi-level persistent homology and electrostatic persistence for chemical and biological characterization, analysis and modeling by using convolutional neural networks. Horn et al. propose a trainable topological layer that incorporates global topological information of a graph using persistent homology. Chen et al. (2022a) propose a time-aware topological deep learning model that captures interactions and encodes encode time-conditioned topological information. However, to the best of our knowledge, PH and TDA have yet to be employed for diffusion models.

**Diffusion generative model.** Deep generative models broadly include Variational Autoencoders (VAEs) , Generative Adversarial Networks (GANs), Normalizing Flows, Energy-Based Models (EBMs) and Diffusion Models. Diffusion models are first introduced in (Sohl-Dickstein et al., 2015) with diffusion implemented in Gaussian and Bernoulli distributions. Recently, Denoising Diffusion models (Ho et al., 2020) are shown to be capable of generating high-dimensional images via the improvement of model architecture and reparametrization of predictions. In (Sohl-Dickstein et al., 2015; Ho et al., 2020), the authors formulated the diffusion process with a discrete Markov chain and reversed the diffusion via a variational probabilistic optimization objective. Nichol & Dhariwal (2021) enhance diffusion models via multiple strategies including avoiding a suboptimal solution resulted from linear noise scheduler and learning variance. Song et al. (2020) showed that diffusion models with discrete Markov chains (Sohl-Dickstein et al., 2015; Ho et al., 2020; Nichol & Dhariwal, 2021) can be generalized into a continuous SDEs process and can be reversed by an ordinary differential equation (ODE) marginally-equivalent. Kingma & Gao (2023) summarized that diffusion model objectives are equal to a weighted integral of the Evidence Lower Bound (ELBO) objectives over different noise levels.Nonetheless, these existing approaches are limited since these diffusion models focus on the diffusion process in Euclidean space. We have observed that the sampling process from Euclidean space in diffusion models may lose topological information and lead to misshapes in generated data (as shown in Figure 4 (e)). Therefore, in this paper, we consider summarizing the diffusion process with structural degrading and reconstruction with topological fingerprints.

## 3 PRELIMINARIES

**Topological Representation Learning** To capture the underlying shape of data, we employ persistent homology (PH) which is a rapidly emerging research subfield at the interface of data sciences, machine learning and algebraic topology (Chazal & Michel, 2017; Otter et al., 2017; Wasserman, 2018). In particular, let $x$ be the observed data (in our later object generation experiments, $x$ is a 2D image or a point cloud). The fundamental idea behind the PH methodology is that the observed data $x$ represent a sample from a metric space and, due to sampling, the underlying unknown geometric and topological structure of this space has been lost. To extract topological information and higher-order interactions in a systematic and efficient manner, we can build an abstract simplicial complex which is a finite collection of simplices that is closed with respect to inclusion of faces. For instance, the Vietoris-Rips (VR) complex (Carlsson, 2009; Zomorodian, 2010) is a prevalent type of simplicial complex utilized in variety of applications, thanks to its straightforward construction and rapid computational implementation. Moreover, as mentioned by Chen et al. (2019), since images consist of pixels, it is ideal for PH to accommodate this inherent representation of images as a grid. Consequently, it is a more feasible option for extracting topological summaries from images (i.e., allowing the usage of more compact data-structures) by applying cubical complexes to grid structures, rather than utilizing simplicial complexes on point clouds. For instance, in Edelsbrunner & Harer (2010), the filtration of complexes given some function on the pixels may deliver a deeper insight intro the intrinsic image properties. Specifically, let $f$ be a filtration function that maps every simplex to the maximum function value of its vertices (in our case the grayscale value) and let $x_\epsilon = f^{-1}(-\infty, \epsilon], \epsilon \in \mathbb{R}$. Setting an increasing sequence of (dis)similarity thresholds $\epsilon$, i.e., $\epsilon_1 < \epsilon_2 < \ldots \epsilon_n$, results in a nested sequence of cubical complexes $x_{\epsilon_1} \subset x_{\epsilon_2} \subset \cdots x_{\epsilon_n}$ which is referred to *lower-star filtered cubical complex* (Edelsbrunner, 2013). As (dis)similarity threshold $\epsilon$ changes, some topological features are born while others disappear. In general, topological features with a shorter lifespan (i.e., life interval) are referred to as *topological noises* and topological features with long lifespan may have a more robust property in $x$. A $k$-dimensional topological feature (or $k$-hole) represent connected components (0-hole), loops (1-hole) and cavities (2-hole). For each $k$-hole, PH records its first appearance (birth) in the filtration sequence, and disappearance (death) in later complexes with a unique pair $(b_\sigma, d_\sigma)$. The most popular topological summary under the PH

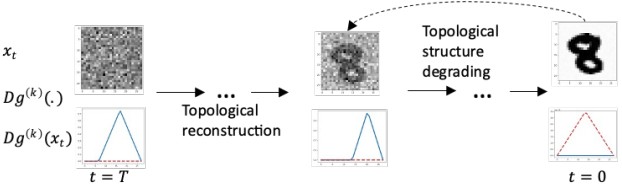

Figure 1: The overall architecture of Topo-Diffusion. Topological structures (i.e., $Dg^{(k)}(x_t)$), which are obtained from TDA transformation $Dg^{(k)}(\cdot)$ from the original data $x_t$, is gradually degraded in the forward process from step $t = 0$ to $t = T$ (denoted as solid arrows). In the reverse process (denoted as dashed arrow), the topological structures are reconstructed via sampling.

framework is a *persistence diagram* (PD). The $k$-th PD is a multi-set of points in a 2D plane, i.e., $Dg^{(k)}(x) = \{(b_i, d_i)^{(k)} \in \mathbb{R}^2 \mid b_i < d_i\}$ that records birth-time ($b$) and death-time ($d$) of each topological feature such as a number of independent components and loops (where $k \in \{0, 1, \dots, \mathcal{K}\}$).

**Diffusion models** Diffusion models (Sohl-Dickstein et al., 2015; Ho et al., 2020) are generative latent variable models with the form of $p_\theta(x_0) = \int p_\theta(x_\theta : T)dx_{1:T}$, where $x_1, \dots, x_T$ are latents of the same dimensionality as the data $x_0$. A diffusion probabilistic model (diffusion model) consists of two steps, i.e., a forward diffusion process and reverse diffusion process. Given a data $x_0$, the forward diffusion process is a parameterized Markov chain to degrade the data distribution by adding noises to a prior distribution (e.g., standard Gaussian) and thus obtaining a variational distribution $q(x_t|x_{t-1})$ over time steps $t \in \{1, 2, \dots, T\}$. That is, the distribution of input data is degraded such that at the final timestep, $x_T$ is degraded to a random noise. The reverse diffusion process is to follow the reverse steps $p(x_{t-1}|x_t) = \mathcal{N}(x_{t-1}; \mu(x_t, t), \sum(x_t, t))$, aiming to generate new samples $p(x_0)$ from the prior distribution $\mathcal{N}(0, \boldsymbol{I})$. Then we employ neural networks to learn $p_\theta(x_{t-1}|x_t) = \mathcal{N}(x_{t-1}; \mu_\theta(x_t, t), \sum \theta(x_t, t))$ that receives as input the noisy image $x_t$ and the embedding at time step $t$, and learns to predict the mean $\mu_\theta(x_t, t)$ and covariance $\sum \theta(x_t, t)$. However, $p_\theta(x_0)$ is intractable as we have to marginalize over all the possible reverse trajectories to compute it. The solution to this problem is to minimize a variational lower-bound of the negative log-likelihood.

# 4 METHODOLOGY

## 4.1 TOPO-DIFFUSION

In this section, we introduce a novel diffusion model, namely, Topo-Diffusion, to enhance performance of classical diffusion models. Topo-Diffusion is capable of (i) utilizing multi-scale topological information of input data and (ii) representing both forward process and reverse process of diffusion model with topological summary . Topo-Diffusion achieves this desirable improvement over existing diffusion models by incorporating topological representation to learn the Markov chain of reverse diffusion process, and reconstructing the data distribution from the prior distribution via a sampling process. The overall architecture of Topo-Diffusion is illustrated in Figure 1. Next, we provide more details about the components of Topo-Diffusion model.

We first introduce the architecture of Topo-Diffusion for topological representation learning. Given a sample $x_t$ at timestep $t$ and a filtration function $f$, we can calculate a set of PDs, i.e., $PH(x_t, f) = \{Dg^{(0)}(x_t), Dg^{(1)}(x_t), \dots, Dg^{(\mathcal{K})}(x_t)\}$ where $\mathcal{K}$ denotes the maximum dimension of homological features of $x_t$. Moreover, it is crucial to pick a class of vectorization and embedding functions that are sufficiently powerful to result in expressive representations of topological signatures. In this study, we consider multiple types of (i) persistence vectorization functions $\phi$ and (ii) differentiable topological embedding function $\Xi(\cdot)$. Formally, the topological diffusion layer can be formulated as

$$a_L^t = \sigma(\Xi([\phi(Dg^{(k)}(x_t))]_{k=0}^{\mathcal{K}}), \tag{1}$$

$$\phi(Dg^{(k)}(x_t)) = \mathcal{F}([w(b_i^t, d_i^t)^{(k)} \cdot s_\theta(b_i^t, d_i^t)^{(k)}]_{(b_i^t, d_i^t)^{(k)} \in Dg^{(k)}(x_t)}), \tag{2}$$

where $[\phi(Dg^{(k)}(x_t))]_{k=0}^{\mathcal{K}} = [\phi(Dg^{(0)}(x_t)), \phi(Dg^{(1)}(x_t)), \dots, \phi(Dg^{(\mathcal{K})}(x_t))]$ denotes a set of vectorized topological features, $[\cdot, \cdot]$ denotes the operation of concatenation, $s_\theta : \mathbb{R}^2 \mapsto \mathbb{R}^q$ denotes

the persistence point transformation, i.e., a mapping function from a persistence point $(b_i^t, d_i^t)^{(k)}$ in the $Dg^{(k)}(x_t)$ to a representation vector, $w$ is the weighting function with respect to the persistence of each persistence point, and $\mathcal{F}$ denotes a permutation-invariant operation such as max or average operation (which is capable of efficiently aggregating high-dimensional representations). We utilize a layer-wise embedding method to learn a latent local topological representation $a_L^t$, and thus improve the reverse process as well as make the topological representation comparable to the dimension of input in the reverse diffusion model. According to Eq. 1, we employ a feed-forward neural network $\Xi(\cdot)$ with a non-linear activation function $\sigma(\cdot)$ to embed the topological representation. Note that, in this study, we consider the embedding function $\Xi(\cdot)$ based on residual learning methods (He et al., 2016b).

Next, we introduce two persistence point transformation methods (see Eq. 2), i.e., (i) birth-lifespan transformation and (ii) differentiable distribution transformation. These transformations can transform a PD (i.e., a multi-set of persistence points) into a topological feature vector or function which can be easily integrated into any type of models. More importantly, the stability property of resulting vectorized topological features such as Betti curves, persistence landscapes, and persistence images is the key aspect that we consider in this paper (see Theorem 4.1 for more details). For the sake of simplicity, in the following, we omit the superscript $(k)$ and treat the $Dg(x_t)$ as the input $k$-dimensional PD for a vectorization transformation.

**Birth-lifespan transformation.** We introduce explicit constructions of topological summaries for persistence vectorizations through the birth-lifespan transformation using *Betti curve* and *persistence landscape*. Specifically, (i) Betti curve allows us to measure various topological information with a given filtration and does not inherit any implicit biases; and (ii) the advantages of persistence landscapes are invertible, stable, and nonlinear.

*Betti curve* is one of the simplest vectorization as it gives the count of topological feature at a given threshold interval. Given a PD and a weight function $\omega : \mathbb{R}^2 \mapsto \mathbb{R}$, the Betti curve of a PD is the function $s_\theta : \mathbb{R}^2 \mapsto \mathbb{R}$ which is defined as

$$\phi_B(Dg(x_t), \tau) = \sum_{(b_i^t, d_i^t) \in Dg(x_t)} \omega(b_i^t, d_i^t) \cdot \mathbb{1}_{[b_i^t, d_i^t]}(\tau),$$

where the persistence point transformation $s_\theta$ is the indicator function $\mathbb{1}_{[b_i^t, d_i^t]}(\tau) = 1$ if $\tau \in [b_i^t, d_i^t]$, otherwise 0, and the permutation-invariant operation $\mathcal{F}$ is a summation.

*Persistence landscape* (PL) is one of the most common vectorizations introduced by Bubenik (2015). For a given persistence diagram $Dg(x_t) = \{(b_i^t, d_i^t)\}$, PL produces a function $\phi(Dg(x_t), \zeta)$ by using generating functions $\Lambda_i$ for each $(b_i^t, d_i^t) \in Dg(x_t)$, i.e., $\Lambda_i : [b_i^t, d_i^t] \mapsto \mathbb{R}$ is a piecewise linear function obtained by two line segments starting from $(b_i^t, 0)$ and $(d_i^t, 0)$ connecting to the same point $(\frac{b_i^t + d_i^t}{2}, \frac{b_i^t - d_i^t}{2})$. Then, the PL function $\phi_{PL}(Dg(x_t), \zeta)$ for $\zeta \in [\epsilon_1, \epsilon_q]$ is defined as

$$\phi_{PL}(Dg(x_t), \zeta) = \max_i \Lambda_i(\zeta),$$

where $\{\epsilon_k\}_1^q$ are thresholds for the filtration used. Considering the piecewise linear structure of the function, $\phi_{PL}(Dg(x_t), \zeta)$ is completely determined by its values at $2q - 1$ points, i.e., $\frac{b_i^t \pm d_i^t}{2} \in \{\epsilon_1, \epsilon_{1.5}, \epsilon_2, \epsilon_{2.5}, \ldots, \epsilon_q\}$ where $\epsilon_{k.5} = (\epsilon_k + \epsilon_{k+1})/2$. Finally, we can obtain a vector of size $1 \times (2q - 1)$, i.e., $\phi_{PL}(Dg(x_t), \zeta) = [\phi_{PL}(Dg(x_t), \epsilon_1), \phi_{PL}(Dg(x_t), \epsilon_{1.5}), \ldots, \phi_{PL}(Dg(x_t), \epsilon_q)]$. Here, the permutation-invariant operation $\mathcal{F}$ is $k$-th largest value, and the weighting function is a constant weight function, i.e., $w(b_i^t, d_i^t) = 1$.

**Differentiable distribution transformation.** Here, we describe the differentiable distribution transformation in Topo-Diffusion, where the basic idea is to capture the location of the persistence points with a differential distribution function. Different from the birth-lifespan transformation, differentiable distribution transformation generates 2D-vectors. From the Definition 4.1, let $s_\theta$ be a differentiable distribution function with $\theta = (\mu, \sigma)$, mean $\mu \in \mathbb{R}^2$ and variance $\sigma \in \mathbb{R}^2$ (e.g., $s_\theta$ can be specified as the Gaussian distribution function $s_\theta(z) = \frac{1}{2\pi\sigma^2} e^{-\frac{||z - u||^2}{2\sigma^2}}$). Note that one may choose a weighting function by using the lifespan of the persistence point (i.e., $w(d_i^t, b_i^t) = d_i^t - b_i^t$) or choose a weighting function that emphasizes persistence points near the death-axis (i.e., $w(d_i^t, b_i^t) = \arctan(C(d_i^t - b_i^t)^2)$ where $C$ is a non-negative parameter). In this setting, one way of vectorization is based on the differentiable distribution transformation such as

Persistence Image (PI) (Adams et al., 2017), which represents a PD as a finite-dimensional vector derived from the weighted kernel density function. [Differentiable Distribution Transformation] Given a (linear) transformed persistence diagram $T(Dg(x_t))$, a persistence surface $\rho_{T(Dg(x_t))} : \mathbb{R}^2 \mapsto \mathbb{R}$ is defined as, for any $z \in \mathbb{R}^2$, $\rho_{T(Dg(x_t))}(z) = \sum_{T(b_i^t, d_i^t) \in T(Dg(x_t))} w(T(b_i^t, d_i^t)) s_\theta(z)$, where $T(b_i^t, d_i^t) = (b_i^t, d_i^t - b_i^t)$.

**Stability of topological summaries.** We now show that the persistence vectorization transformation in Topo-Diffusion is stable (for the proof of the Theorem 4.1, please refer to Appendix D).

**Theorem 4.1.** *Given two observed datasets $x_t$ and $x_{t'}$, we have the stability equation as follows*

$$d(\phi(Dg(x_t), \phi(Dg(x_{t'}))) \leq \mathcal{C}_\phi \cdot \mathcal{W}_{p_\phi}(Dg(x_t)), Dg(x_{t'})),$$

*where $d(\cdot, \cdot)$ is a suitable metric on the space of a persistence vectorization, $\phi(Dg(x_t))$ and $\phi(Dg(x_{t'}))$ represent the vectorizations for $Dg(x_t)$ and $Dg(x_{t'})$ respectively, $\mathcal{W}_p$ is Wasserstein-p distance, and $1 \leq p_\phi \leq \infty$.*

Specifically, if a given persistence vectorization holds the stability inequality as below for some $d$ and $\mathcal{W}_p$, we call the persistence vectorization is a *stable* vectorization in terms of Wasserstein-p distance. Hence, our proposed Topo-Diffusion can (i) improve the stability of the local topological features learning, (ii) increase the robustness of both deterministic forward and reverse processes against errors and noises, and (iii) consequently enhance the quality of generated samples.

As such, in this work, we develop a denoising decoder with residue blocks to learn the reverse Markov chain with topological features. In each layer $a_L^t$ of the denoising decoder, we employ the backbone (Ho et al., 2020) in which timestep embeddings $S^t$ and topological representation embeddings can be embedded, and the output of layer $a_L^t$ is represented as

$$a_e^t = [\phi(Dg^{(k)}(x_t))]_{k=0}^{\mathcal{K}} \cdot a_L^t + S^t. \tag{3}$$

By stacking backbones in a U-Net architecture, we build a practically beneficial reverse topological diffusion model.

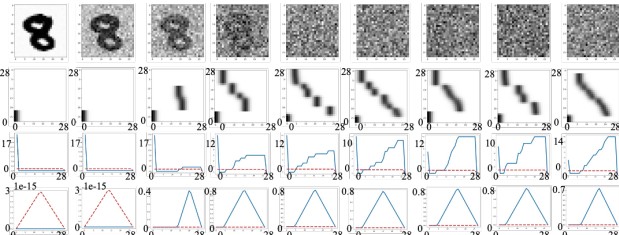

Figure 2: An illustration of vectorized topological features at diffusion steps in the forward process. From top to bottom: images, persistence images, Betti curves, persistence landscapes. In Betti curves and persistence landscapes, blue lines and red dash lines represent the 0-dimensional and 1-dimensional topological features respectively. From left to right: objects in the forward process with diffusion steps $t = \{0, 20, 30, 50, 100, 150, 200, 250, 300\}$.

### 4.2 FORWARD PROCESS

**Topological structure degrading.** The forward process of diffusion model refers to gradually corrupt data with noises (Kingma et al., 2021; Nichol & Dhariwal, 2021; Ho et al., 2020). In Topo-Diffusion, we consider the corruption process in a topological aspect via a topological structure degrading. Let $p(x_0)$ be the data density, where the index 0 represents the original data, we gradually add noise to the topological structure with a Markov chain $x_1, x_2, ..., x_T$ are obtained according to the following Markovian process:

$$q(\mathcal{C}(x_t, [\phi(Dg^{(k)}(x_t))]_{k=0}^{\mathcal{K}})|x_{t-1}) := \mathcal{N}(x_t; \sqrt{1 - \beta_t} x_{t-1}), \beta_t \boldsymbol{I}), \forall t \in \{1, ..., T\}, \tag{4}$$

where $T$ denotes the total number of diffusion steps, $\beta_1, \beta_2, \ldots, \beta_T$ are variances scheduler to degrade the original data, $\boldsymbol{I}$ is the identity matrix with the same dimensions as $x_0$, $\mathcal{N}(x; \mu, \sigma)$ is a

normal distribution of mean $\mu$ and covariance $\sigma$ that produces $x$. As mentioned in Ho et al. (2020), the forward process admits sampling $x_t$ at an arbitrary timestep $t$ in a closed form: using the notation $\alpha_t := 1 - \beta_t$ and $\bar{\alpha} := \prod_{s=1}^{t} \alpha_s$, we have

$$q(\mathcal{C}(x_t, [\phi(Dg^{(k)}(x_t))]_{k=0}^{\mathcal{K}})|x_0) = \mathcal{N}(x_t; \sqrt{\bar{\alpha}_t}x_0, (1 - \bar{\alpha}_t\boldsymbol{I})). \tag{5}$$

In this study, we set $\beta_1, \beta_2, \ldots, \beta_T$ as hyperparameters so that the forward diffusion process is not included in training.

### 4.3 Reverse process

**Topological reconstruction.** By leveraging properties of the forward process, we can generate new samples from $p(x_0)$ by sampling from $x_T := \mathcal{N}(0, \boldsymbol{I})$ in the reverse process. Figure 3 shows the learning procedure of the reverse process of Topo-Diffusion. Instead of reversing from the disturbed data itself, we incorporate topological representation of the degraded data into the posterior of forward process as $p_\theta(x_{0:T})$

$$p_\theta(x_{0:T}) := p(\mathcal{C}(x_T, [\phi(Dg^{(k)}(x_T))]_{k=0}^{\mathcal{K}}) \times \prod_{t=1}^{T} p_\theta(x_{t-1}|\mathcal{C}(x_t, [\phi(Dg^{(k)}(x_t))]_{k=0}^{\mathcal{K}})),$$

$$p_\theta(x_{t-1}|\mathcal{C}(x_t, [\phi(Dg^{(k)}(x_t))]_{k=0}^{\mathcal{K}})) := \mathcal{N}(x_{t-1}; \mu_\theta(\mathcal{C}(x_t, [\phi(Dg^{(k)}(x_t))]_{k=0}^{\mathcal{K}}), t), \tag{6}$$

$$\sum_\theta(\mathcal{C}(x_t, [\phi(Dg^{(k)}(x_t))]_{k=0}^{\mathcal{K}}), t)).$$

With the tractable forward process $q(x_{t-1}|\mathcal{C}(x_t, [\phi(Dg^{(k)}(x_t))]_{k=0}^{\mathcal{K}})$ under the condition of $x_0$

$$q(x_{t-1}|\mathcal{C}(x_t, [\phi(Dg^{(k)}(x_t))]_{k=0}^{\mathcal{K}}, x_0) = \mathcal{N}(x_{t-1}; \tilde{\mu}_t(\mathcal{C}(x_t, [\phi(Dg^{(k)}(x_t))]_{k=0}^{\mathcal{K}}, x_0), \tilde{\beta}_t\boldsymbol{I})$$

$$\tilde{\mu}_t(\mathcal{C}(x_t, [\phi(Dg^{(k)}(x_t))]_{k=0}^{\mathcal{K}}, x_0) := \frac{\sqrt{\bar{\alpha}_{t-1}}\beta_t}{1 - \bar{\alpha}_{t-1}}x_0 + \frac{\sqrt{\bar{\alpha}_t}(1 - \bar{\alpha}_{t-1})\mathcal{C}(x_t, [\phi(Dg^{(k)}(x_t))]_{k=0}^{\mathcal{K}}}{1 - \bar{\alpha}_t} \tag{7}$$

$$\tilde{\beta}_t = \frac{1 - \bar{\alpha}_{t-1}}{1 - \bar{\alpha}_t}\beta_t$$

The training process of the reverse diffusion model is performed by optimizing the usual variational bound on a negative log-likelihood (NLL)

$$\mathbb{E}[-log p_\theta(x_0)] \leq \mathbb{E}_q[-log\frac{p_\theta(x_{0:T})}{q(x_{1:T}|x_0)}]$$

$$= \mathbb{E}_q[-log p(x_T) - \sum_{t\geq 1} log\frac{p_\theta(x_{t-1}|\mathcal{C}(x_t, [\phi(Dg^{(k)}(x_t))))]_{k=0}^{\mathcal{K}}))}{q(\mathcal{C}(x_t, [\phi(Dg^{(k)}(x_t))]_{k=0}^{\mathcal{K}}|x_{t-1})}]. \tag{8}$$

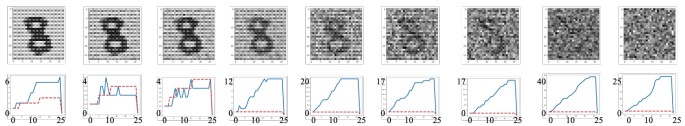

Figure 3: *Top:* the reverse process of the Topo-Diffusion model trained by Betti curves. *Bottom:* the reconstruction of Betti curves during sampling. From left to right: objects in the reverse process with sampling steps $t = \{0, 20, 30, 50, 100, 150, 200, 250, 300\}$.

## 5 Experiments

**Data sets.** We evaluate our proposed Topo-Diffusion model on a suite of experiments using three simulated point cloud datasets and four real-world image datasets (see Appendix B for more details). The three point cloud datasets are toy examples with different topological structures including generated circles, s-curves, and moon curves. In addition, for real-world image

Table 1: Comparison of coupling generative models on point cloud data and image data with our proposed Topo-Diffusion model across multiple evaluation metrics. PL: persistence landscape. BC: Betti curve. PI: persistence image. The best results are highlighted in bold.

| | | | Point clouds | | | Images | | |
|---|---|---|---|---|---|---|---|---|
| | | | Circles | S-curve | moon curve | MNIST | FashionMNIST | LFW |
| FID score | Topo-Diffusion | PL | 0.2587 | 0.2250 | 5.8625 | 2.5729 | **0.1861** | 9.2861 |
| | | BC | **0.2138** | **0.2221** | **5.8411** | **2.2799** | 0.2304 | **7.8730** |
| | | PI | 0.2304 | 0.2986 | 5.8693 | 3.0242 | 0.5831 | 11.1763 |
| | Baseline | | 0.2737 | 0.3916 | 6.5864 | 3.0461 | 0.3493 | 13.1552 |
| Precision | Topo-Diffusion | PL | 0.8494 | 0.7398 | 0.6201 | **0.7530** | **0.8790** | 0.7330 |
| | | BC | **0.8756** | **0.7832** | **0.6780** | 0.7040 | 0.7670 | **0.8640** |
| | | PI | 0.8544 | 0.5586 | 0.6691 | 0.7024 | 0.4570 | 0.5860 |
| | Baseline | | 0.8324 | 0.3930 | 0.5963 | 0.3200 | 0.6650 | 0.5950 |
| Recall | Topo-Diffusion | PL | 0.7890 | 0.7038 | 0.8018 | **0.8500** | **0.8330** | 0.4810 |
| | | BC | **0.8203** | **0.7576** | **0.8594** | 0.8370 | 0.7660 | **0.9730** |
| | | PI | 0.8025 | 0.7555 | 0.7937 | 0.7808 | 0.8060 | 0.6350 |
| | Baseline | | 0.7789 | 0.5586 | 0.7793 | 0.4985 | 0.7280 | 0.7330 |
| RMSE | Topo-Diffusion | PL | 1.3223 | 1.2871 | 2.7907 | **0.1117** | **0.1236** | 3.3477 |
| | | BC | 1.2726 | **1.2451** | **2.7892** | 0.1159 | 0.1276 | **2.9744** |
| | | PI | **1.1274** | 1.2800 | 2.8064 | 0.1283 | 0.1568 | 3.6590 |
| | Baseline | | 1.3311 | 1.2725 | 2.9336 | 0.1298 | 0.1680 | 4.6011 |
| SSIM | Topo-Diffusion | PL | 0.1600 | 0.0206 | 0.0161 | **0.0190** | 0.0088 | 0.0096 |
| | | BC | **0.1680** | **0.0208** | **0.0194** | 0.0184 | **0.0090** | **0.0165** |
| | | PI | 0.1630 | 0.0196 | 0.0101 | 0.0164 | 0.0079 | 0.0102 |
| | Baseline | | 0.1580 | 0.0050 | 0.0093 | 0.0174 | 0.0086 | 0.0046 |

datasets, we use MNIST (Deng, 2012), Fashion MNIST (Xiao et al., 2017b), Labelled Face in the Wild (LFW) (Huang et al., 2007), and Circuit Reconstruction from Electron Microscopy Images (CREMI) (CRE).

**Experimental settings.** In the experiments, we ensure that reverse and forward processes have approximately the same functional form while keeping the signal-to-noise ratio at $x_T$ as small as possible ($L_T = D_{KL}(q(x_T|x_0)|N(0, I)) \approx 10^{-5}$). We measure the quality and diversity of generated samples utilizing multiple metrics including Frechet Inception Distance (FID), precision and recall (Kynkäänniemi et al., 2019b), as well as the similarity between real and generated data using Root Mean Square Error (RMSE) and Similarity Index (SSIM) (Kynkäänniemi et al., 2019b; Sajjadi et al., 2018b) (See Appendix C for more details). For the reverse process, we employ a U-Net (Ronneberger et al., 2015) based on residue learning (Zagoruyko & Komodakis, 2016). Our U-Net maps features into the resolution of 4 for 1-D data and $4 \times 4$ for 2-D images. All models have two convolutional residual blocks per resolution level and self-attention blocks at the $16 \times 16$ resolution between the convolutional blocks. Diffusion time $t$ is specified by adding the Transformer sinusoidal position embedding into each residual block. The topological feature is embedded to every residual block via Eq. 3.

**Results.** First, we assess the fidelity and diversity of generated data using Topo-Diffusion evaluated using multiple metrics. Table 1 summarizes the results of our Topo-Diffusion on three point-cloud patterns and three real-world image datasets. Our results clearly show that the incorporation of topological features in diffusion model can largely improve the quality and diversity of generated data, compared to the baseline diffusion model (Dhariwal & Nichol, 2021; Nichol & Dhariwal, 2021). For instance, on the Circles dataset, our Topo-Diffusion (based on BC) achieves the best performance across all five evaluation metrics, and has an average 12.40% relative gain, compared with the baseline. On the three image datasets, Topo-Diffusion (based on PL) outperforms the baseline with an average relative gain 49.25%, 33.56%, 35.45%, 29.86%, and 20.93% on FID score, precision, recall, RMSE, and SSIM respectively. Second, we conduct ablation experiments to evaluate the generated results by varying parameters or strategies implemented in Topo-Diffusion. Specifically, we train the Topo-Diffusion on various sets of parameters for PL and PI on MNIST dataset, and select parameters of PI/PL with the best FID score. We found that the optimal topological feature varies among different datasets. Furthermore, we observe that our Topo-Diffusion based on Betti curve outperforms the DDPM diffusion (Dhariwal & Nichol, 2021) model on all datasets. As shown in Figure 4, we can clearly observe that a misshape of point cloud occurs in the baseline model; while

the structure of "S" curve of the training data is maintained in the generated point cloud by utilizing our Topo-Diffusion model. This justifies the benefit of the integration of topological features into the diffusion model which can help lead towards a both stable and discriminative result. Finally, we conduct ablation studies to evaluate the effect of topological embeddings on the performance of Topo-Diffusion (discussed in Section 4.1). As summarized in Table 2, comparing with the quantitative evaluation metrics on generated Fasion-MNIST samples from embedded/non-embedded Topo-Duffision models, we oberve that both the fidelity and diversity of generated samples are enhanced by the embedding approach of Topo-Diffusion. These results indicate that topological embeddings proposed in Topo-Diffusion are critical in the diffusion process of diffusion models.

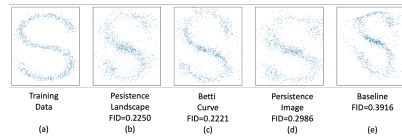

Figure 4: "S" curve point cloud from (a) the training dataset, and generated samples from (b) Topo-Diffusion based on persistence landscapes, (c) Topo-Diffusion based on Betti curves, (d) Topo-Diffusion based on persistence image, and (e) DDPM Diffusion model.

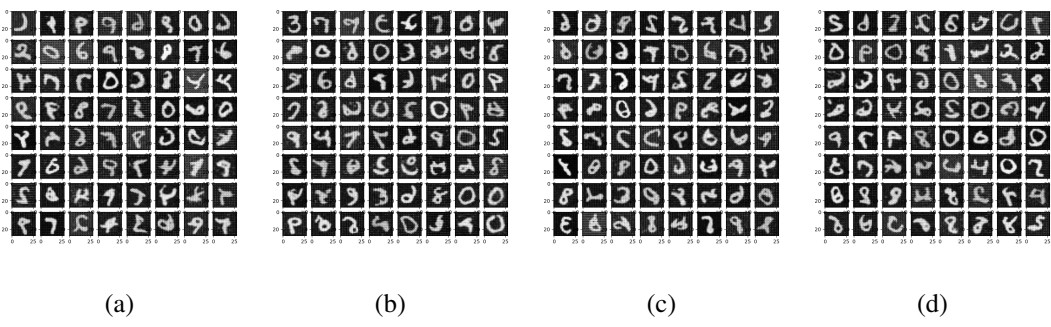

Figure 5: Comparison between generated MNIST samples from (a) baseline (DDPM), (b) Topo-Diffusion based on persistence landscape, (c) Topo-Diffusion based on Betti curve, and (d) Topo-Diffusion based on persistence image.

Table 2: The qualities of generated Fasion-MNIST samples in Topo-diffusion with/without TDA feature embedding.

|  | FID | Precision | Recall | RMSE | SSIM |
|---|---|---|---|---|---|
| Emb. PL | 0.1861 | 0.8790 | 0.8330 | 0.1236 | 0.0088 |
| Emb. BC | 0.2304 | 0.7670 | 0.7660 | 0.1276 | 0.0090 |
| Unemb. PL | 0.2087 | 0.8440 | 0.8270 | 1.3070 | 0.0086 |
| Unemb. BC | 0.2316 | 0.7540 | 0.7630 | 1.3361 | 0.0067 |

## 6    CONCLUSION

In this study, we have explored the utility of local topological fingerprints to enhance various data generation tasks within the diffusion model paradigm. We propose a novel topological diffusion model called Topo-Diffusion that incorporates persistent homology and topological representation learning into diffusion models. We derive theoretical stability guarantees of the proposed topological features of generated data. Through extensive empirical evaluations, we demonstrate that our Topo-Diffusion can enhance generating point clouds data, structural images, and can also yield state-of-the-art performances on real-world image datasets. In the future, we plan to investigate the capability of diffusion models in fusing single persistence and multipersistence topological features (Carriere & Blumberg, 2020; Chen et al., 2022b). Moreover, to alleviate the impact of topological noise and yield a faster approximation of high-dimensional topological representations, we will apply the witness complex (De Silva & Carlsson, 2004) to our current Topo-Diffusion architecture. We will also extend Topo-Diffusion to model diverse data types across various application domains.

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

APPENDICES OF TOPO-DIFFUSION: TOPOLOGICAL DIFFUSION MODEL FOR IMAGE AND POINT CLOUD GENERATION

## A THE TOPO-DIFFUSION MODEL

### A.1 DETAILS OF THE MODEL

Based on the backbone of PixelCNN++ (Salimans et al., 2017; Dhariwal & Nichol, 2021), we modify the ResNet-based U-Net with 1D convolution kernel for point cloud data. We map all input into $4 \times 4$ resolution as previous works did (Dhariwal & Nichol, 2021; Nichol & Dhariwal, 2021), with two residual block in each resolution level in both baseline model and our Topo-diffusion model. Our point cloud models have 0.44 million parameters and our $28 \times 28$ models have 13 million parameters and $64 \times 64$ models have 100 million parameters. We train a large model which have 4 residual block, aiming to learn the topological information and time embedding respectively. We utilize NVIDIA GPU A5800 for all experiments. For point clouds, we extract the topological features at 0.95, 0.17, 0.06 seconds per sample respectively by using persistence image, persistence landscape, and Betti curve. For image data, we extract the topological features at 8.95, 4.49, 4.46 seconds per sample respectively by using persistence image, persistence landscape, and Betti curve. Moreover, we train on point cloud data for 0.23M steps, MNIST and Fashion MNIST for 2.4M steps, LFW for 1.8M steps respectively. Due to the memory constraints, we conduct the hyperparameters search to optimize for circle pattern point cloud and MNIST datasets.

- We choose the $\beta_t$ schedule from linear (Dhariwal & Nichol, 2021), cosine (Nichol & Dhariwal, 2021; Dhariwal & Nichol, 2021), quadratic (Dhariwal & Nichol, 2021) with $T = [10, 50, 100, 200, 300, 1000]$, and we choose the cosine scheduler with $T = 10$ and $T = 300$ for point cloud and image data respectively, which is identity with previous work (Nichol & Dhariwal, 2021).

- We train our Topo-Diffusion model both *with pretraining network* and *without pretraining network*. In the pertraining, we set all vectorized topological summaries as identity vectors. Thus there is no affect from topological features shown in Eq. (3) (i.e., $a_t^T = 1$). In our experiments, we found the pretraining network improve sample quality and converge speed for all datasets.

- We train our Topo-Diffusion model with Adam and SGD optimizer. We set the learning rate to be $1e - 4$. In addition, we warm up the network by setting the learning rate as $1e - 6$ with 100 steps on image datasets. In our experiments, we found the warm up on image datasets improve the stability of training thus enhance the sample quality for all datasets.

Final experiments are trained once and evaluated throughout the training procedure for the sample quality in FID score. Sample quality scores and log likelihood are reported on the final model.

### A.2 DETAILS OF THE ALGORITHM

Here we provide more details about the algorithm of our Topo-Diffusion model. Algorithm 2 represents the sampling procedure from random noise $N(0, 1)$ with $\mathcal{C}(x_t, [\phi(Dg^{(k)}(x_t))]_{k=0}^{\mathcal{K}}$ as learnt topological information. To generate reasonable samples, it is up to capture the dependencies of topological reconstruction along the reverse process. While sampling with reconstructed topological is straightforward, the difficulties lie in the optimization of Topo-Diffusion model. Algorithm 1 represents the process of each round of training. We train the embedding network $\phi(\cdot)$ with denoising autoencoder simultaneously by optimizing Eq. 8. We utilize the co-train approach to make training efficiently by employing same time embedding method in Dhariwal & Nichol (2021).

## B DATA

### B.1 DATASETS

We evaluate our proposed Topo-Diffusion model on three 2-dimensional (2D) point cloud, and three real-world image datasets. The three simulated point cloud datasets are circles, s-curves, and moon

---

**Algorithm 1** Training process of Topo-Diffusion

---

**Require:** $x_0, \phi(\cdot), Dg(\cdot)$
1: $t \sim Uniform(1, \dots, T)$
2: $\epsilon \sim N(0, 1)$
3: take gradient from
4: $E_{x_0, [\phi(Dg^{(k)}(x_0))]_{k=0}^{\mathcal{K}}, \epsilon}[||\epsilon - \epsilon_\theta(x_t, [\phi(Dg^{(k)}(x_t))]_{k=0}^{\mathcal{K}}, \epsilon)||^2]$
5: **until** converge

---

**Algorithm 2** Sampling from topological reconstruction

---

**Require:** $T, \tilde{\beta}_1, \dots, \tilde{\beta}_T$
1: $x_T \sim N(0, 1)$
2: **for do**$t = T, \dots, 1$ do
3:     **if** $t > 1$ **then**
4:        $z \sim N(0, 1)$
5:     **else**
6:        $z = 0$
7:     $\tilde{\mu}_t = \frac{\sqrt{\bar{\alpha}_{t-1}}\beta_t}{1-\bar{\alpha}_{t-1}}x_0 + \frac{\sqrt{\alpha_t}(1-\bar{\alpha}_{t-1})\mathcal{C}(x_t, [\phi(Dg^{(k)}(x_t))]_{k=0}^{\mathcal{K}}}{1-\bar{\alpha}_t}$
8:     $x_{t-1} = \tilde{\mu}_t + \tilde{\beta}_t \cdot z$
9:     **Return** $x_0$

---

curves from scikit-learn tools (Pedregosa et al., 2011). The circle dataset is a synthetic large circle containing a smaller circle where their ratio of radius is $2 : 1$. The moon point clouds (Eq. 9), s-curve point cloud (Eq. 10) are generated using trigonometric functions. All three point clouds are perturbed by the standard Gaussian noise.

$$
\begin{aligned}
\boldsymbol{m}_1 &= (cos(\boldsymbol{x}),\ sin(\boldsymbol{x})) \\
\boldsymbol{m}_2 &= (1 - cos(\boldsymbol{x}),\ 1 - sin(\boldsymbol{x}) - 0.5)
\end{aligned} \tag{9}
$$

$$
\boldsymbol{s} = (sin(\boldsymbol{x}), sign(\boldsymbol{x}) \times (cos(\boldsymbol{x}) - 1)) \tag{10}
$$

where $\boldsymbol{x}$ follows a uniform distribution from 0 to $\pi$. For each kind of 2D point clouds, we generate 10,000 training samples and each sample consists of 1,024 points.

Image datasets includes MNIST ($28 \times 28$ greyscale handwritten digits) (Deng, 2012), Fashion MNIST ($28 \times 28$ greyscale Zalando's article images) (Xiao et al., 2017a), and Labelled Face in the Wild (LFW) (greyscale human faces images) (Huang et al., 2007). For MNIST and Fashion MNIST, we randomly select a set of 10,000 samples as a training set. LFW data consists of 13,233 samples, and we select the region $((61, 189), (61, 189))$ of each image (where the main face are included), and then map the sliced images to the size of $64 \times 64$.

### B.2 Data preprocessing

To ensure the neural network reverse process operates on consistently scaled inputs starting from the standard normal prior $p(x_t)$, We preprocess data by linearly normalizing the input into the scaling of $[-1, 1]$.

## C Experiment Results

**Experimental settings.** For forward process, we employ cosine noise scheduler with variance from $\beta_1 = 0.0001$ to $\beta_T = 0.02$ to diffuse the original samples to random noise. We set $T = 10$ and $T = 300$ to 2D point clouds and image datasets respectively, ensuring that reverse and forward processes have approximately the same functional form while keeping the signal-to-noise ratio at $x_T$ as small as possible ($L_T = D_{KL}(q(x_T|x_0)|N(0, I)) \approx 10^{-5}$).

For reverse process, we employ a U-Net (Ronneberger et al., 2015) based on a wide ResNet (Zagoruyko & Komodakis, 2016). Our U-Net map the feature into resolution of 4 for 1D

data and $4 \times 4$ for 2D images. All models have two convolutional residual blocks per resolution level and self-attention blocks at the $16 \times 16$ resolution between the convolutional blocks. Diffusion time $t$ is specified by adding the Transformer sinusoidal position embedding into each residual block. The topological feature is embeded to the residual block in the first layer of the U-Net.

## C.1 METRICS

**Frechet Inception Distance.** The Frechet Inception Distance (FID) score (Heusel et al., 2017) is a popularly utilized statistics metric that calculates an 1D distance score between feature vectors of real and generated images. Lower scores indicate the two groups of images are more similar.

**Pecision and Recall.** We have evaluated sample fidelity and diversity by widely used precision and recall metrics, which are calculated via the distance to the third nearest neighbor in a latent feature space (Kynkäänniemi et al., 2019a; Sajjadi et al., 2018a). In the precision and recall metrics, high precision means high generative fidelity and high recall represents high generative diversity. In this paper, we empirically select $k = 10$.

**Root Mean Squared Error.** We have employed root mean squared error (RMSE) (rms, 2008) to quantify the difference of points or pixels between the generated sample and the training set of the Topo-Diffusion model, as a measurement of sample quality. Specifically, lower RMSE values represent higher similarities between the generated sample and the reference sample (training set).

**Structural Similarity Index.** In this paper, we have adopted a structure similarity index (SSIM) metric Wang et al. (2004) to evaluate the generated sample quality. The SSIM assesses the structural information based on the degradation of structural features Wang et al. (2004). For the SSIM, high value represents higher similarity between two group of samples. For all point clouds and images datasets, we calculate the global SSIM to measure the similarities between the generated data and the reference data (i.e., training set). For point clouds, we use an 1D convolution kernel with Gaussian weighing as a local window which moves point-by-point over the point clouds aiming at calculating local statistics and SSIM index. For point clouds, we use an 2D convolution kernel with Gaussian weighing as a local window which moves pixel-by-pixel over the image to calculate local statistics and SSIM index.

**Embedding Model for Evaluation.** Since FID Heusel et al. (2017), precision and recall Kynkäänniemi et al. (2019a); Sajjadi et al. (2018a), and RMSE rms (2008) are usually conducted on an embedded space, we train classifiers between each point cloud pattern and random Gaussian noise ($\mu = 0$, $\sigma = 1$), respectively, with a 3-layer multi-layer perceptron. We take the latent vector before the classification output head (i.e., a 10-D vector), for evaluation. For image datasets, we train ResNets He et al. (2016a) with 3 blocks for evaluation. For MNIST and Fashion MNIST datasets, we use the classification labels from each dataset as labels for the classifier. For LFW dataset, we select those people with over 70 images in the dataset as training set of the ResNet and use the name of human face samples as labels for classifier. For all three image datasets, we take the latent vector before the classification output head for calculating each evaluation score, in which a 50-D vector is used.

## C.2 ADDITIONAL RESULTS

**Enhancing density estimation.** We estimate the density of each component using 2D-GMM, and visualize densities of simulated and generated modes in 3D plots (see Figures 6, 7, 9, 10, 12, and 13). We found that the proposed Topo-Diffusion model enhances densities estimation over all point cloud patterns. For baseline DDPM model, a sampled point cloud centered around the origin point in corordinates, which is one of the local optimals. We evaluate the Topo-Diffusion model in aspect of density estimation, which indicates the topological information enhances the model for learning the density distribution. The results show that data generated from Topo-diffusion outperformed the DDPM, which indicates that including topological feature in diffusion process enhance the sampling process of diffusion model.

**Comparison with more generative models** We train our Topo-Diffusion model with a pretraining network. We set all vectorized topological summaries as identity vectors in the pertraining and pretrain the Topo-Diffusion model for 100k steps. After pretraining, we train all of the models

(including baseline or DDPM, Topo-Diffusion with persistence landscape, Topo-Diffusion with betti curves, Topo-Diffusion with persistence image) for 1 million steps (Figures 15-17).

Moreover, we have conducted extensive experiments and experimental evaluations show that our proposed Topo-Diffusion always outperforms other generative models (including GAN, WGAN and TopoGAN) on MNIST and CREMI datasets (Tables 3 and 4). As shown in Figure 19, we randomly sampled 64 images from (a) CREMI dataset; (b) Topo-Diffusion model, (c) DDPM, (d) Topo-GAN, and (e) WGAN. We sampled the CREMI dataset into sizes of $64 \times 64$. The visualization shows that a diffusion based model outperformed a GAN based model in the diversity phase. Furthermore, topological aware methods (i.e., Topodiffusion and TopoGAN) can capture more topological mode of the original dataset. From qualitative visualizations, we can see that the image quality as well as diversity can be improved from the TopoDiffusion model.

Table 3: Comparison of the performance of Topo-Diffusion and baseline methods on CREMI.

|  | FID | Precision | Recall | RMSE |
|---|---|---|---|---|
| WGAN | 23.014 | 0.657 | 0.351 | 6.650 |
| TopoGAN | 20.495 | 0.837 | 0.635 | 6.520 |
| DDPM | 21.373 | 0.765 | 0.842 | 6.780 |
| **Topo-Diffusion (ours)** | **18.450** | **0.886** | **0.856** | **6.450** |

Table 4: Comparison of the performance of Topo-Diffusion and baseline methods on MNIST.

|  | FID | Precision | Recall | RMSE |
|---|---|---|---|---|
| WGAN | 3.125 | 0.516 | 0.580 | 0.184 |
| TopoGAN | 3.124 | 0.694 | 0.615 | 0.177 |
| DDPM | 3.046 | 0.320 | 0.499 | 0.130 |
| **Topo-Diffusion (ours)** | **2.280** | **0.704** | **0.837** | **0.116** |

Table 5: Comparison of the performance of Topo-Diffusion and baseline methods on NASA satellites images.

|  | FID | Precision | Recall | RMSE |
|---|---|---|---|---|
| WGAN | 31.020 | 0.694 | 0.484 | 37.893 |
| DDPM | 10.796 | 0.789 | 0.875 | 13.959 |
| **Topo-Diffusion (ours)** | **6.521** | **0.869** | **0.891** | **7.909** |

**Combinations of topological features and transformations** We have presented additional experimental results, i.e., combinations of topological features and transformations in TopoDiffusion including 1) loops and connected components of betti curve, loops and connected components of persistence landscape (BC+PL); 2) loops and connected components of betti curve (CC+Loop); 3) loops of betti curve (Loop); 4) connected components of betti curve (CC) (see Tables 6, 7, and 8)

Table 6: Comparison of the performance of different combinations of topological features and transformations in TopoDiffusion on $S$ curve.

|  | FID | Precision | Recall | RMSE | SSIM |
|---|---|---|---|---|---|
| BC+PL | 0.751 | 0.718 | 0.669 | 0.439 | 0.053 |
| CC+Loop | 0.908 | 0.668 | 0.666 | 0.478 | 0.034 |
| Loop | 1.101 | 0.616 | 0.666 | 0.539 | 0.003 |
| CC | 1.077 | 0.662 | 0.551 | 0.501 | 0.001 |

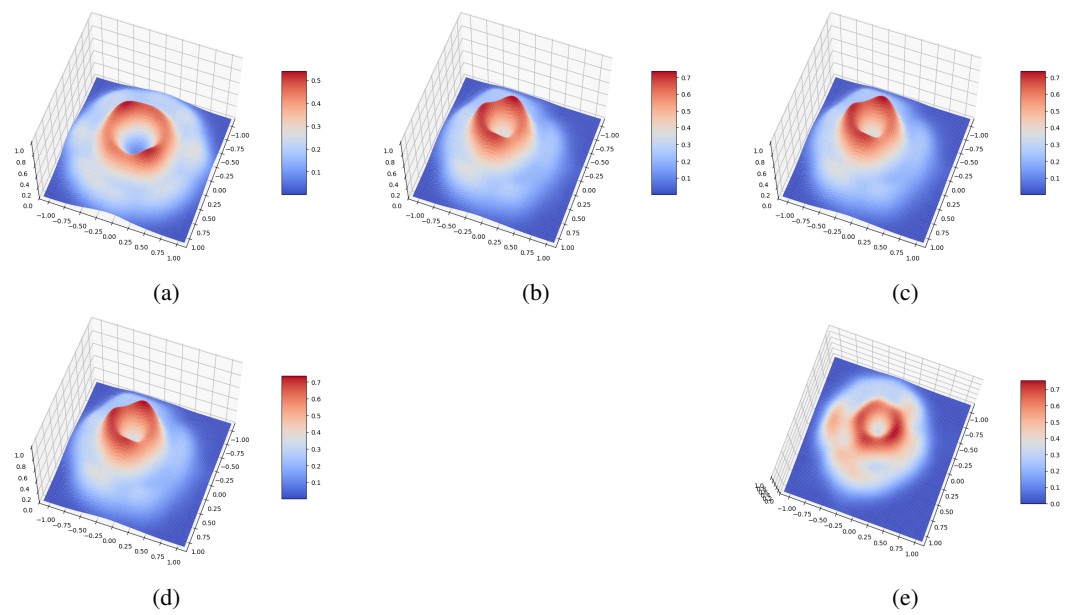

Figure 6: Comparison of density estimation between real and generated s-curves from our Topo-Diffusion and the baseline. (a) Real data, (b) Topo-Diffusion based on Betti curve, (c) Topo-Diffusion based on persistence landscape, (d) Topo-Diffusion based on persistence image, and (e) baseline (DDPM).

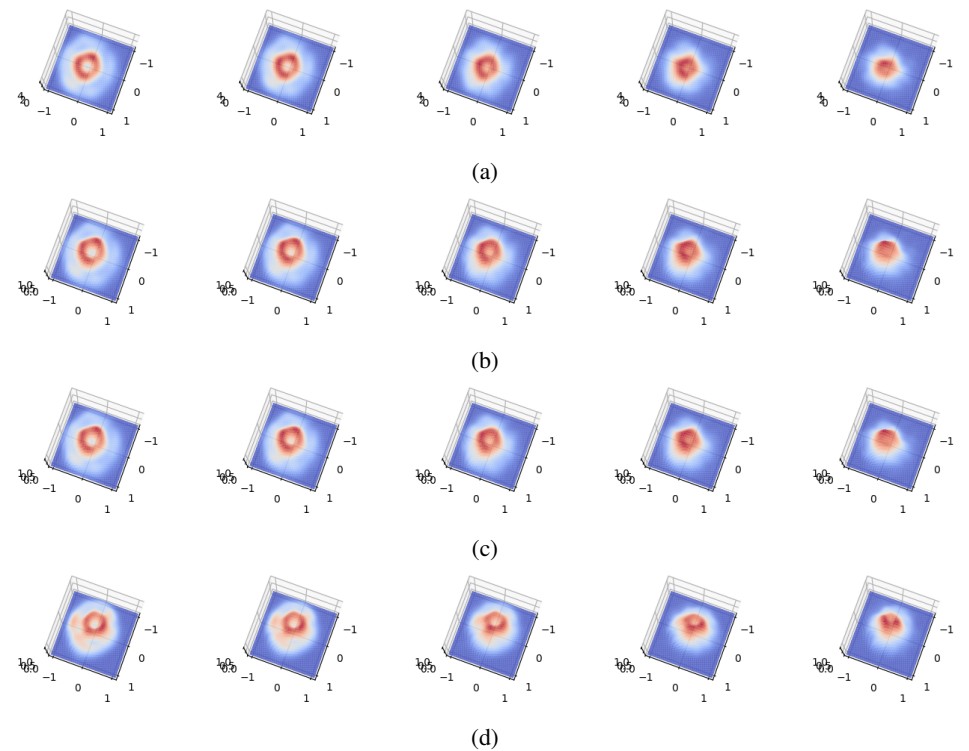

Figure 7: Comparison of density estimation of sampling procedure between our Topo-Diffusion and the baseline at $t = \{2, 4, 6, 8, 10\}$ from right to left. (a) Topo-Diffusion based on Betti curve, (b) Topo-Diffusion based on persistence landscape, (c) Topo-Diffusion based on persistence image, and (d) baseline (DDPM).

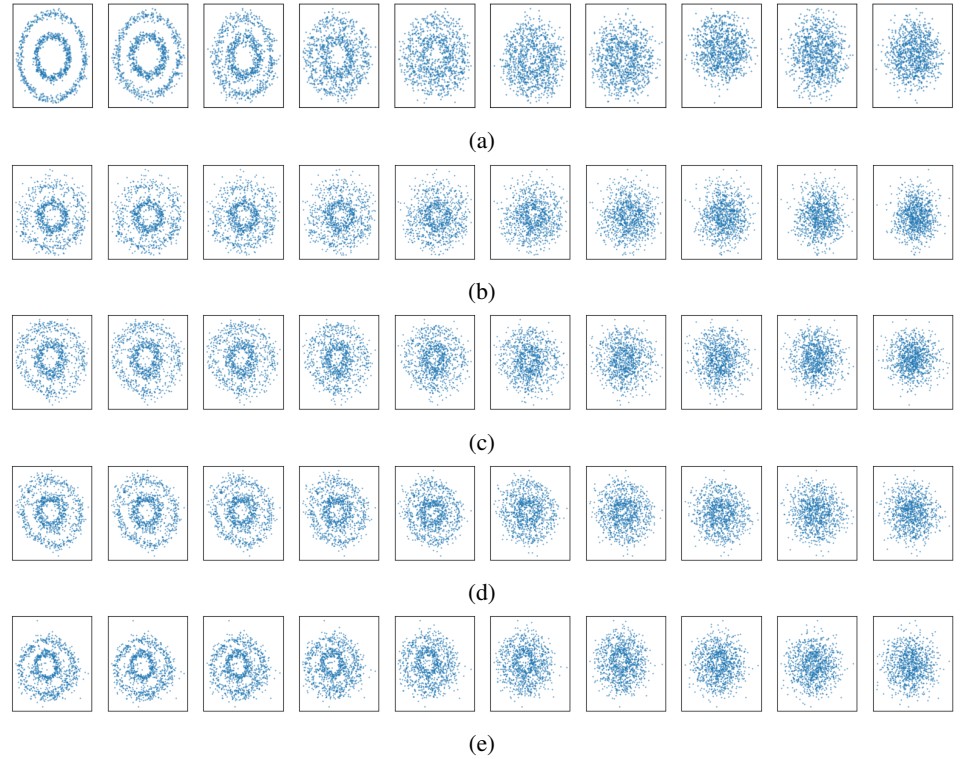

Figure 8: Comparison between real and generated data at $t = \{1, 2, 3, 4, 5, 6, 7, 8, 9, 10\}$ from right to left. (a) Real data, (b) Topo-Diffusion based on Betti curve, (c) Topo-Diffusion based on persistence landscape, (d) Topo-Diffusion based on persistence image, and (e) baseline (DDPM).

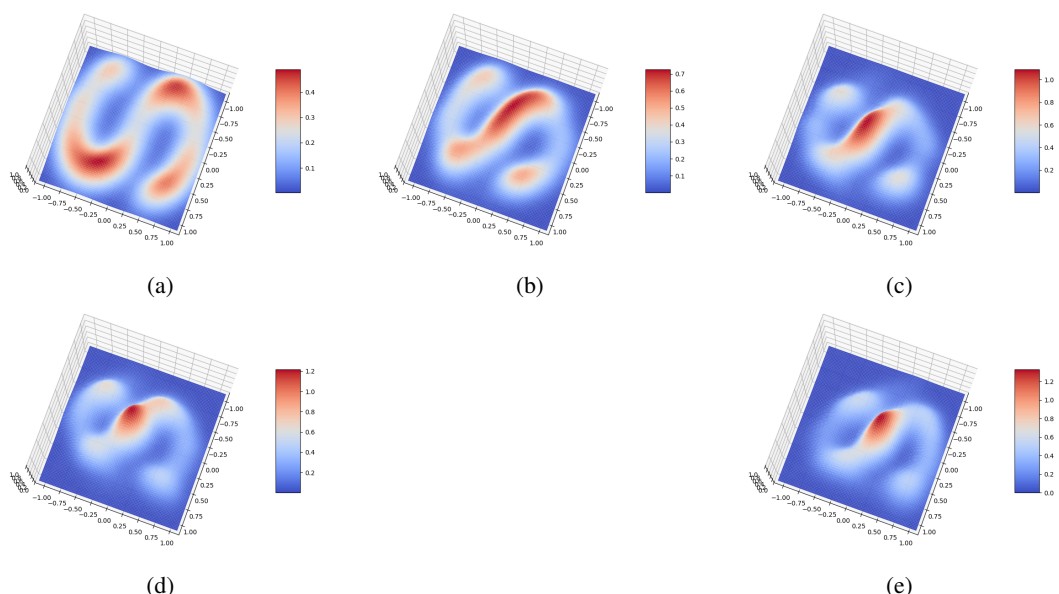

Figure 9: Comparison of density estimation between real and generated s-curves by our Topo-Diffusion and the baseline. (a) Real data, (b) Topo-Diffusion based on Betti curve, (c) Topo-Diffusion based on persistence landscape, (d) Topo-Diffusion based on persistence image, and (e) baseline (DDPM).

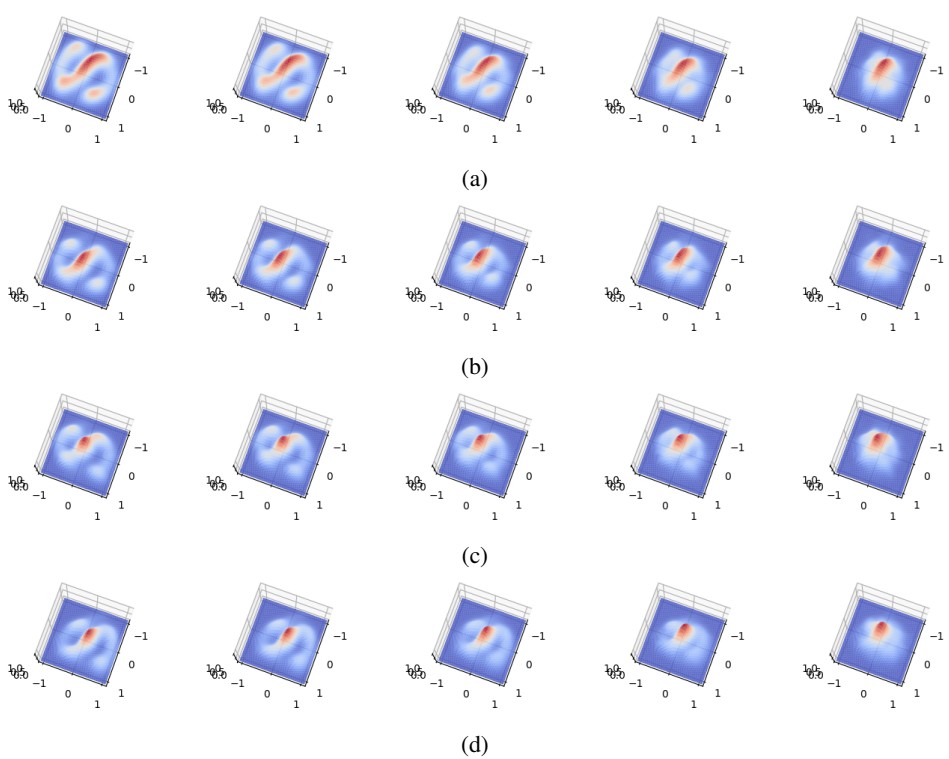

Figure 10: Comparison of density estimation of sampling procedure between our Topo-Diffusion and the baseline at $t = \{2, 4, 6, 8, 10\}$ from right to left. (a) Topo-Diffusion based on Betti curve, (b) Topo-Diffusion based on persistence landscape, (c) Topo-Diffusion based on persistence image, and (d) baseline (DDPM).

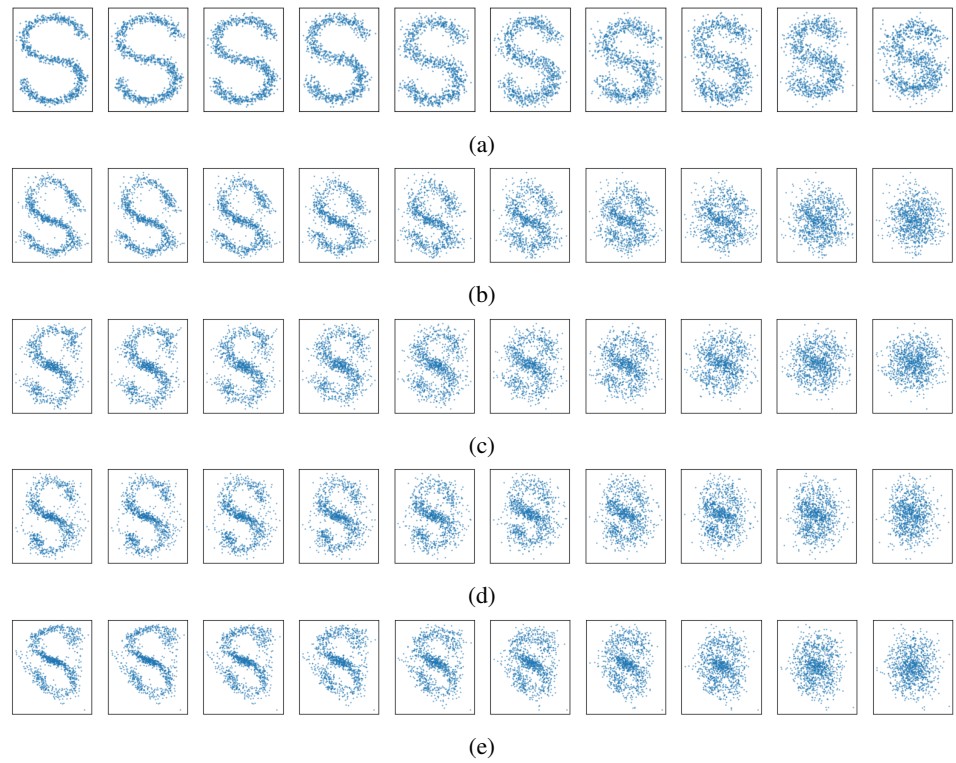

Figure 11: Comparison between real and generated data at $t = \{1, 2, 3, 4, 5, 6, 7, 8, 9, 10\}$ from right to left.. (a) Real data, (b) Topo-Diffusion based on Betti curve, (c) Topo-Diffusion based on persistence landscape, (d) Topo-Diffusion based on persistence image, and (e) baseline (DDPM).

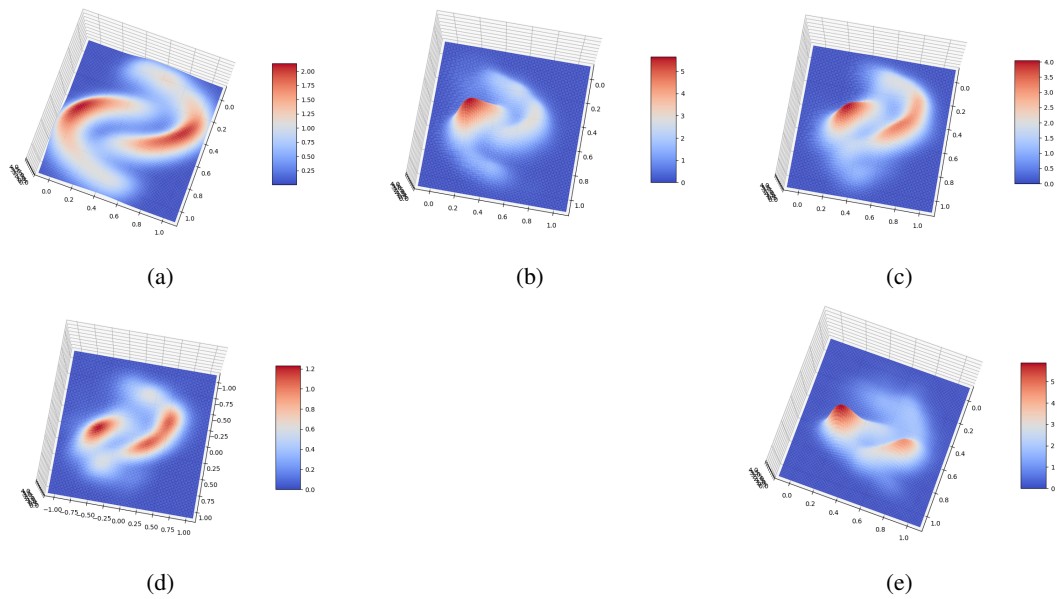

Figure 12: Comparison of density estimation of moon data between real and generated s-curves by our Topo-Diffusion and baseline. (a) Real data, (b) Topo-Diffusion based on Betti curve, (c) Topo-Diffusion based on persistence landscape, (d) Topo-Diffusion based on persistence image, and (e) baseline (DDPM).

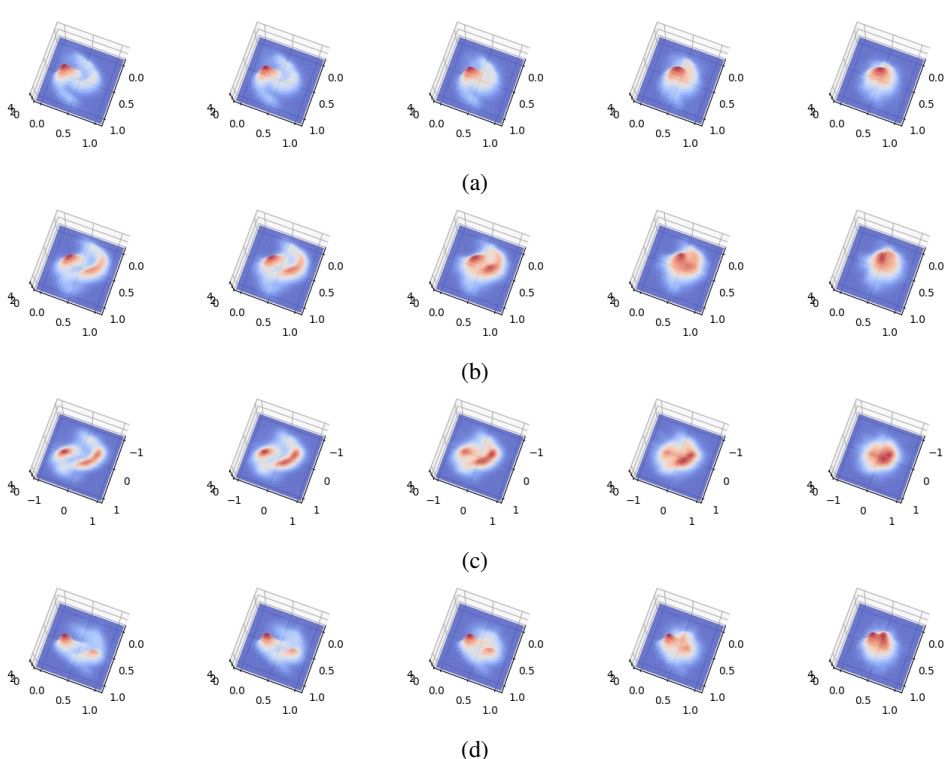

Figure 13: Comparison of density estimation of sampling procedure of moon data between our Topo-Diffusion and the baseline at $t = \{2, 4, 6, 8, 10\}$ from right to left. (a) Topo-Diffusion based on Betti curve, (b) Topo-Diffusion based on persistence landscape, (c) Topo-Diffusion based on persistence image, and (d) baseline (DDPM).

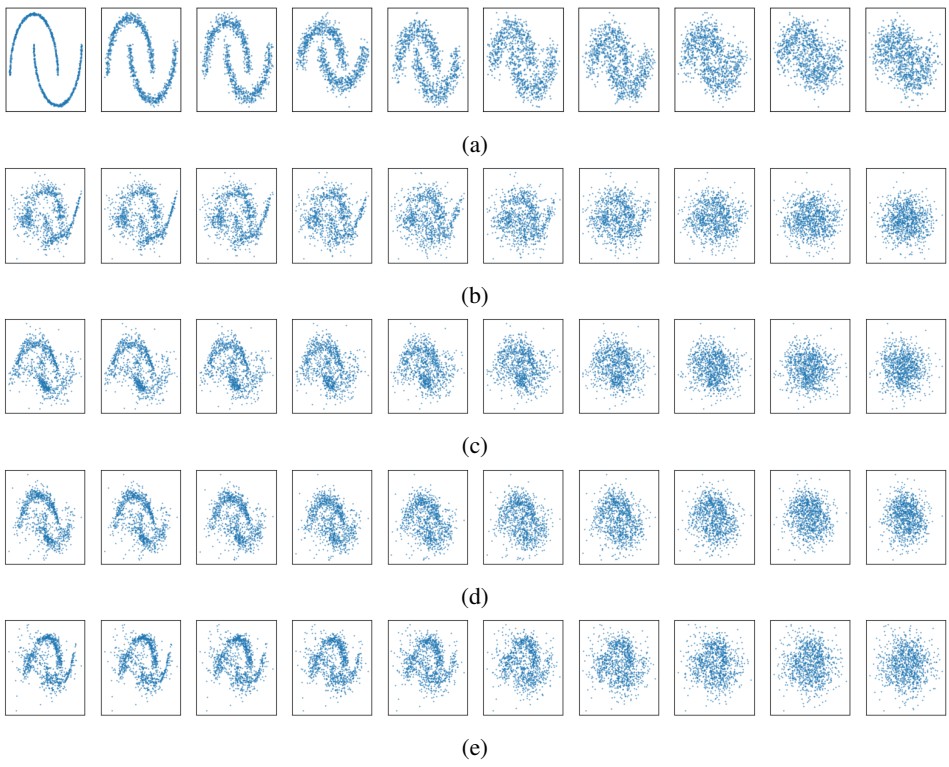

(a)

(b)

(c)

(d)

(e)

Figure 14: Comparison between real and generated moon data at $t = \{1, 2, 3, 4, 5, 6, 7, 8, 9, 10\}$ from right to left. (a) real data, (b) Topo-Diffusion based on Betti curve, (c) Topo-Diffusion based on persistence landscape, (d) Topo-Diffusion based on persistence image, and (e) baseline (DDPM).

Table 7: Comparison of the performance of different combinations of topological features and transformations in TopoDiffusion on Moon.

|         | FID   | Precision | Recall | RMSE  | SSIM  |
|---------|-------|-----------|--------|-------|-------|
| BC+PL   | 1.840 | 0.798     | 0.648  | 0.600 | 0.013 |
| CC+Loop | 1.867 | 0.755     | 0.535  | 0.601 | 0.010 |
| Loop    | 1.961 | 0.633     | 0.407  | 0.612 | 0.006 |
| CC      | 2.052 | 0.529     | 0.362  | 0.639 | 0.006 |

Table 8: Comparison of the performance of different combinations of topological features and transformations in TopoDiffusion on Circle.

|         | FID   | Precision | Recall | RMSE  | SSIM  |
|---------|-------|-----------|--------|-------|-------|
| BC+PL   | 0.088 | 0.873     | 0.665  | 0.265 | 0.195 |
| CC+Loop | 0.117 | 0.529     | 0.708  | 0.280 | 0.166 |
| Loop    | 0.148 | 0.484     | 0.664  | 0.291 | 0.146 |
| CC      | 0.164 | 0.478     | 0.610  | 0.302 | 0.143 |

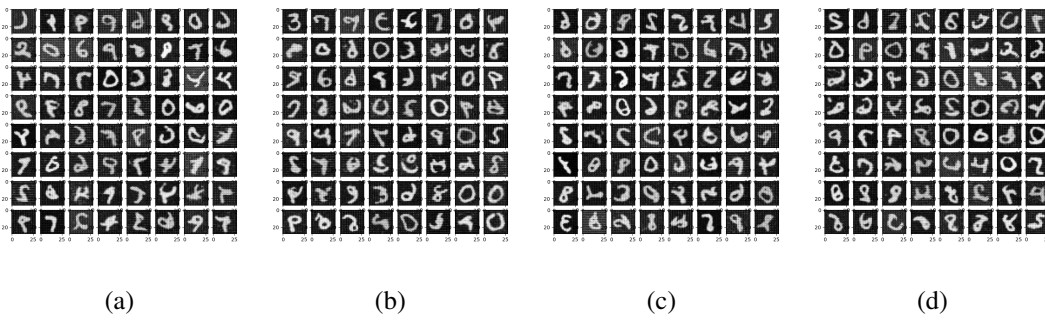

|  (a)  |  (b)  |  (c)  |  (d)  |

Figure 15: Comparison between generated MNIST samples from (a) baseline (DDPM), (b) Topo-Diffusion based on persistence landscape, (c) Topo-Diffusion based on Betti curve, and (d) Topo-Diffusion based on persistence image.

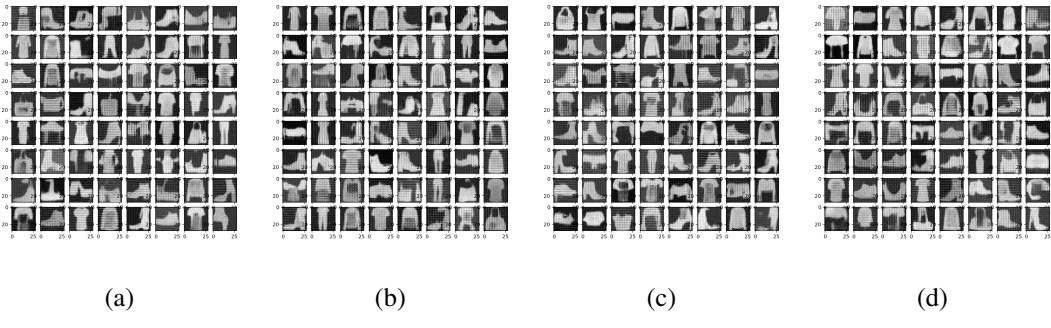

|  (a)  |  (b)  |  (c)  |  (d)  |

Figure 16: Comparison between generated Fashion MNIST samples from (a) baseline (DDPM), (b) Topo-Diffusion based on persistence landscape, (c) Topo-Diffusion based on Betti curve, and (d) Topo-Diffusion based on persistence image.

## D  THEORETICAL ANALYSIS

We provide the details concerning the proof of Theorem 4.1 as follows.

*Proof.* Let $Dg(x_t) = \{q_j^t\} \cup \Delta_t$ and $Dg(x_{t'}) = \{q_j^{t'}\} \cup \Delta_{t'}$ where $\Delta_t$ and $\Delta_{t'}$ represent the diagonal with infinite multiplicity of $Dg(x_t)$ and $Dg(x_{t'})$ respectively, and $q_j^t$ and $q_j^{t'}$ represent the

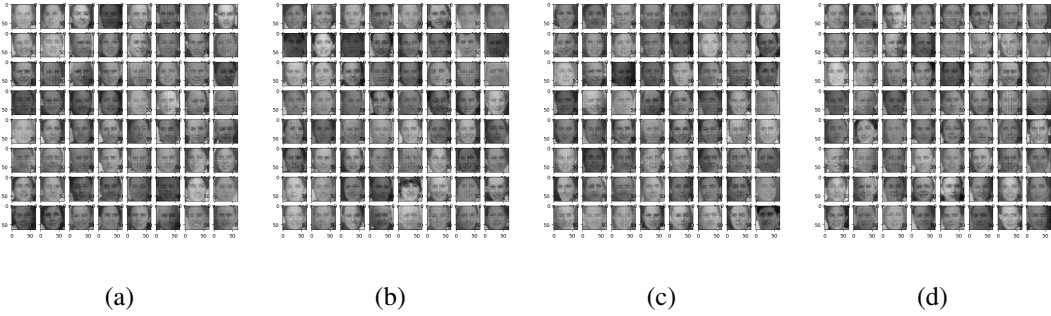

(a)          (b)          (c)          (d)

Figure 17: Comparison between generated LFW samples from (a) baseline (DDPM), (b) Topo-Diffusion based on persistence landscape, (c) Topo-Diffusion based on Betti curve, and (d) Topo-Diffusion based on persistence image.

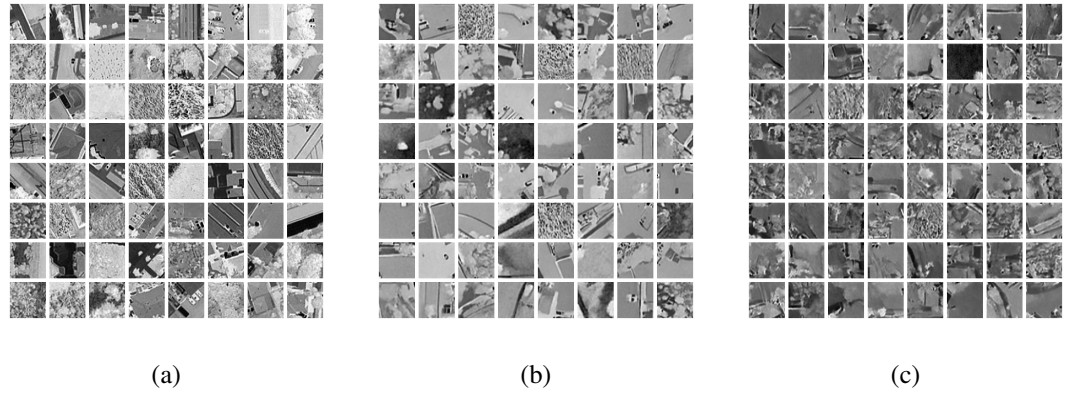

(a)                    (b)                    (c)

Figure 18: Visualization of (a) Training data, and generated NASA satellites images from (b) Topo-diffusion (betti curve), and (c) DDPM.

birth and death times a hole $\sigma_j$ in $x_t$ and $x_{t'}$ respectively. Let $\Upsilon : Dg(x_t) \mapsto Dg(x_t)$ represent a bijective matching. Then the Wasserstein-$p$ distance can be defined as

$$\mathcal{W}_p(Dg(x_t), Dg(x_t)) = \min_{\Upsilon}(\sum_j ||q_j^t - \Upsilon(q_j^t)||_\infty^p)^{\frac{1}{p}}, \quad p \in \mathbb{Z}^+.$$

A persistence vectorization $\phi(Dg(x_t))$ is stable if $d(\phi(Dg(x_t)), \phi(Dg(x_{t'}))) \leq \mathcal{C}_\phi \cdot \mathcal{W}_{p_\phi}(Dg(x_t), Dg(x_{t'}))$, and the constant $\mathcal{C}_\phi > 0$ is independent of $x_t$ and $x_{t'}$. □

### D.1 TOPOLOGICAL SUMMARIZATION

Figure 20 depicts an example of barcode representations (including 0-, 1-, and 2-dimensional topological features, i.e., $H_0$, $H_1$, and $H_2$) of homology over a point cloud. We observe that, by gradually changing the threshold $\epsilon$, the barcode representation can filter out topological noises, and capture significant topological and higher-order features.

In brief, the key idea here is to choose some suitable scale parameters $\epsilon$ to study changes in homology that occur to $x$ which evolves with respect to $\epsilon$. That is, we no longer treat $x$ as a single object but as a *filtration* $x_{\epsilon_1} \subseteq \ldots \subseteq x_{\epsilon_n} = x$, induced by monotonic changes of $\epsilon$. To make the process of pattern counting more systematic and efficient, we build an abstract simplicial complex $\mathscr{K}(x_{\alpha_j})$ on each $x_{\alpha_j}$, resulting in a filtration of complexes $\mathscr{K}(x_{\epsilon_1}) \subseteq \ldots \subseteq \mathscr{K}(x_{\epsilon_n})$. For instance, we can select a scale parameter as a distance (e.g., edge weight) between any two points; then generate an abstract simplicial complex $\mathscr{K}(x_{\epsilon_*})$ by producing sub-point clouds $x'$ with a bounded diameter $\epsilon_*$ (i.e., $(k-1)$-simplex in $\mathscr{K}(x_{\epsilon_*})$ is made up by sub-point clouds $x'$ of $k$-nodes with $diam(x') \leq \epsilon_*$).

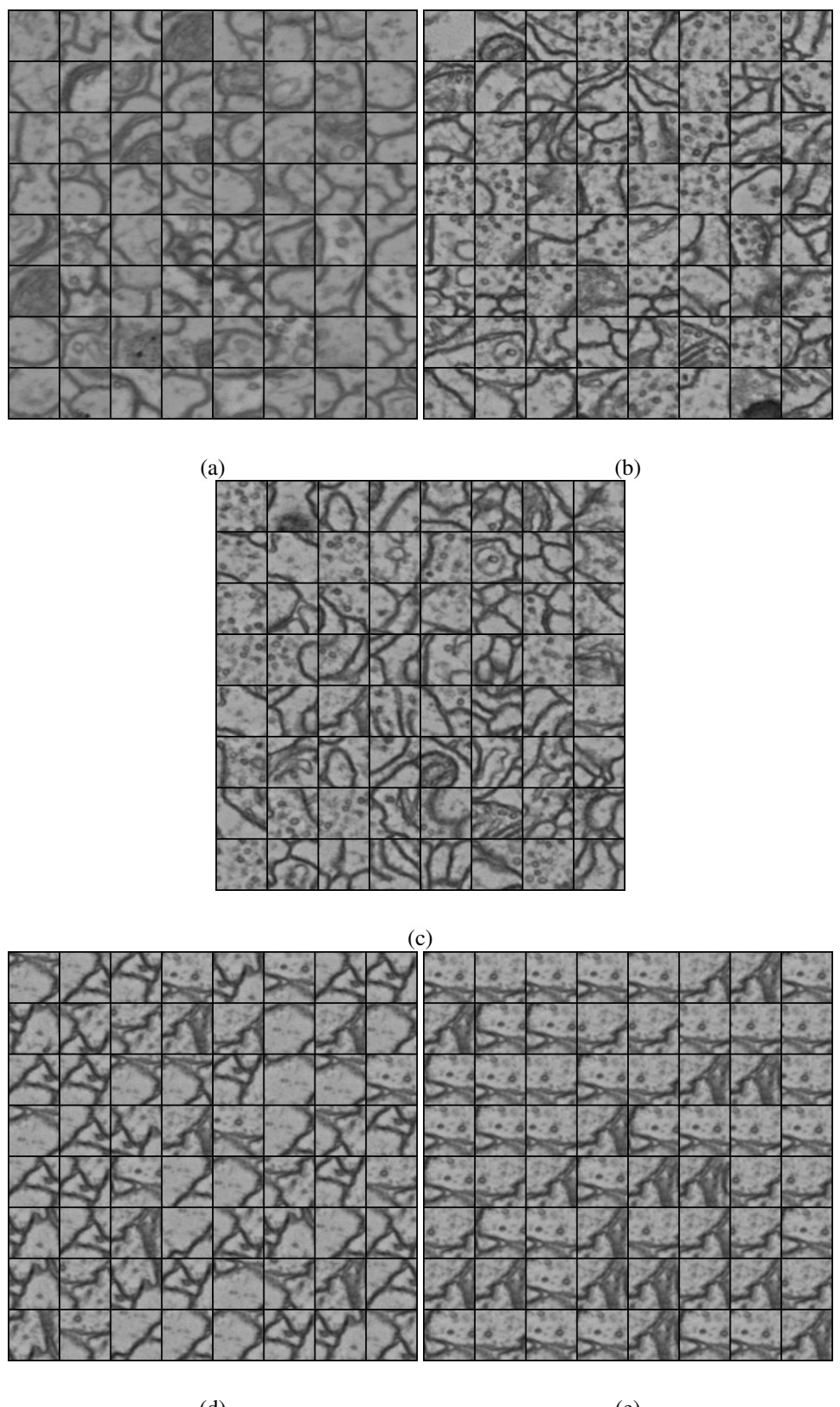

(a)                 (b)

(c)

(d)                 (e)

Figure 19: Visualization of (a) Training data, and generated CREMI images from (b) Topo-Diffision model, (c) DDPM, (d) Topo-GAN, and (e) WGAN.

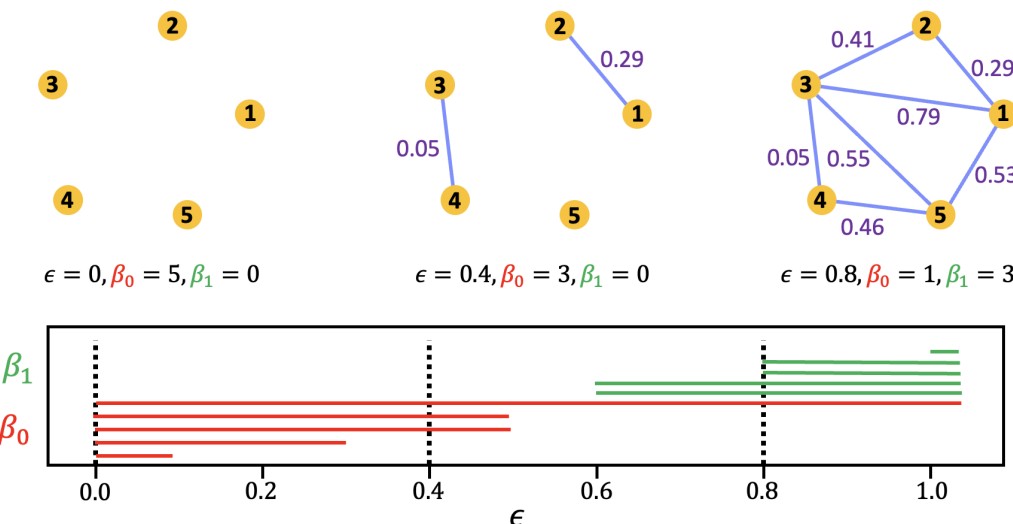

Figure 20: The persistent barcode of one-dimensional Vietoris-Rips filtration built over five points. The top three subfigures are snapshots of the evolving complex as threshold $\epsilon$ increases. Note that the positions of the points in the figure do not exactly reflect the interpoint distances. The distances between two points in the point cloud less than or equal to $\epsilon$ are depicted in blue. The $\epsilon$ values corresponding to the two ends of the horizontal bars mark the birth and death times of topological features. We can observe that (i) when $\epsilon = 0$, $\beta_0 = 0$ (i.e., $H_0$) and $\beta_1 = 0$ (i.e., $H_1$); (ii) when $\epsilon = 0.4$, $\beta_0 = 3$ and $\beta_1 = 0$; and (iii) when $\epsilon = 0.8$, $\beta_0 = 1$ and $\beta_1 = 3$.

## E  UNDERSTANDING TOPO-DIFFUSION WITH ELBO

In our work, the training of topo-diffusion model is optimized using a noise prediction loss. In this regard, topo-diffusion is parameterized as a noise-prediction ($\epsilon$-prediction) model Kingma & Gao (2023); Ho et al. (2020): $s_\theta(x_t, Tp(x_t); \lambda)$, where $x_t$ is degraded data in forward process, $Tp(x_t) = [\phi(Dg^{(k)}(x_t))]_{k=0}^{\mathcal{K}}$ is the topological summaries of $x_t$, $\lambda_t$ is noise scheduler. As the topological summaries are remained stable along the diffusion forward process, for our Topo-Diffusion model, the optimization objective can be simplified as

$$||s_\theta(x_t, Tp(x_t); \lambda) - \nabla log q(x_t|x_0)|| \le \sigma_\lambda^{-2}||\hat{\epsilon}_\theta(x_t, Tp(x_t); \lambda_t) - \epsilon||_2^2, \qquad (11)$$

where $x_0$ is the original data, $\sigma_\lambda$ denotes a parameter associated with noise scheduler. The weighting function of topo-diffusion model is $w(\lambda) \le \sigma_\lambda^{-2}$, which can be treated as a monotonically increasing function of $t$. Thus, we can qualitatively conclude that by incorporating topological summaries into diffusion mdoel, objective of our topo-diffusion model equals to a weighted integral of Evidence Lower Bound Objective (ELBO). In addiction, Denoising Diffusion Probabilistic Model (DDPM) is a noise-prediction based model which does not consider any topological information in the degrading and reconstructing process. The objective of DDPM is $\sigma_\lambda^{-2}||\hat{\epsilon}_\theta(x_t; \lambda_t) - \epsilon||_2^2$, where $w_{DDPM}(\lambda) = \sigma_\lambda^{-2}$ Ho et al. (2020); Kingma & Gao (2023). Comparing Topo-Diffusion and DDPM, we have $w(\lambda) \le w_{DDPM}(\lambda)$. Therefore, incorporating topological information enhances the diffusion process by tightening the weighting function.

**Remark:** In this section, we qualitatively analyze the effects of topological information in Topo-Diffusion model on ELBO-based approximation, compared to DDPM (a noise prediction model without topological information). We remain the quantitative calculation of the weighting function $w(\lambda)$ of Topo-Diffusion to our future work.

*Proof.* In the proposed Topo-Diffusion, we gradually degrade the original data and its topological information using a Markov chain as

$$q(x_t, Tp(x_t)|x_{t-1}) := \mathcal{N}(x_t; \sqrt{1 - \beta_t}x_{t-1}, \beta_t \boldsymbol{I}), \ \forall t \in \{1, ..., T\}, \qquad (12)$$

where $\beta_1, \beta_2, \ldots, \beta_T$ are variances scheduler to degrade the original data, $\boldsymbol{I}$ is the identity matrix with the same dimensions as $x_0$, $\mathcal{N}(x_t; \mu, \sigma)$ is a normal distribution of mean $\mu$ and covariance $\sigma$ that produces $x_t$. With this Markov chain, we can further have a tractable forward process as:

$$
\begin{aligned}
q(x_{t-1}|x_t, Tp(x_t), x_0) &:= \mathcal{N}(x_{t-1}; \tilde{\mu}_t(x_t, Tp(x_t), x_0), \tilde{\beta}_t \boldsymbol{I}), \\
\tilde{\mu}_t(x_t, Tp(x_t), x_0) &:= \frac{\sqrt{\bar{\alpha}_{t-1}}\beta_t}{1-\bar{\alpha}_{t-1}}x_0 + \mathcal{C}(\frac{\sqrt{\alpha}_t(1-\bar{\alpha}_{t-1})}{1-\alpha_t}x_t, Tp(\frac{\sqrt{\alpha}_t(1-\bar{\alpha}_{t-1})}{1-\alpha_t}x_t)), \\
\tilde{\beta}_t &:= \frac{1-\bar{\alpha}_{t-1}}{1-\bar{\alpha}_t}\beta_t,
\end{aligned}
\tag{13}
$$

where $\mathcal{C}$ is an embedding of $x_t$ and $Tp(x_t)$ in our Topo-Diffusion model. To generate samples from $x_T \sim \mathcal{N}(0, 1)$, we train the proposed Topo-Diffusion model to learn the reverse process as follows

$$
\begin{aligned}
p_\theta(x_{0:T}) &:= p(x_T, Tp(x_T) \times \prod_{t=1}^{T} p_\theta(x_{t-1}|x_t, Tp(x_t)), \\
p_\theta(x_{t-1}|x_t, Tp(x_t)) &:= \mathcal{N}(x_{t-1}; \mu_\theta(x_{t-1}|x_t, Tp(x_t)), \sigma_\theta(x_{t-1}|x_t, Tp(x_t)).
\end{aligned}
\tag{14}
$$

With Eq. 12)-(14, the KL divergence $L(t; x_0)$ between the joint distribution of forward process $q(x_{0:T}|x_0)$ and reverse model $p(x_{0:T})$ can be expressed as follows

$$
L(t; x_0) := D_{KL}(q(x_{0:T}|x_0)||p(x_{0:T})). \tag{15}
$$

We denote $dt$ as an infinitesimal change in time. In our work, $L(t; x_0)$ can be decomposed as the sum of KL divergence and an expected KL divergence as

$$
L(t-dt; x_0) = L(t; x_0) + \mathbb{E}_{q(x_t|x_0)}[D_{KL}(q(x_{t-dt}|(x_t, Tp(x_t)), x_0)||p(x_{t-dt}|(x_t, Tp(x_t)))], \tag{16}
$$

Using Eq. 16, time derivative of $L(t; x_0)$ can be expressed as

$$
\begin{aligned}
\frac{d}{dt}L(t; x_0) &= \frac{1}{dt}(L(t; x_0) - L(t-dt; x_0)) \\
&= -\frac{1}{dt}\mathbb{E}_{q(x_t|x_0)}[D_{KL}(q(x_{t-dt}|(x_t, Tp(x_t)), x_0)||p(x_{t-dt}|(x_t, Tp(x_t)))],
\end{aligned}
\tag{17}
$$

where given a diffusion step $t$ and original data $x_0$, $q(x_{t-dt}|(x_t, Tp(x_t)), x_0)$ is tractable by Eq. 13. As we design the reverse process Eq. 14, we can further calculate that as follows

$$
\begin{aligned}
&\mathbb{E}_{q(x_t|x_0)}[D_{KL}(q(x_{t-dt}|x_t, Tp(x_t), x_0)||p(x_{t-dt}|x_t, Tp(x_t)))] \\
&= \mathbb{E}_{q(x_t|x_0)}[\frac{1}{2\sigma_t^2}||\tilde{\mu}_t(x_t, Tp(x_t), x_0) - \mu_\theta(x_t, Tp(x_t), x_0)||_2^2] + c
\end{aligned}
\tag{18}
$$

where $c$ is a constant. In our work, we have developed a model with its prediction $\mu_\theta$ to approximate $\tilde{\mu}_t$, in order to learn the distribution of forward process. Substituting Eq. 12, we can get $x_t = \sqrt{\bar{\alpha}_t}x_0 + \sqrt{1-\bar{\alpha}_t}\epsilon$, $\epsilon \sim \mathcal{N}(0, 1)$, and then Eq. 18 can be expanded as

$$
\begin{aligned}
&\mathbb{E}_{q(x_t|x_0)}[D_{KL}(q(x_{t-dt}|x_t, Tp(x_t), x_0)||p(x_{t-dt}|x_t, Tp(x_t)))] - c \\
&= \mathbb{E}_{x_0, \epsilon}[\frac{1}{2\sigma_t^2}||\tilde{\mu}_t(x_t, Tp(x_t), \frac{1}{\sqrt{\bar{\alpha}_t}}(x_t - \sqrt{1-\bar{\alpha}_t}\epsilon) - \mu_\theta(x_t, Tp(x_t), x_0)||_2^2] \\
&= \mathbb{E}_{x_0, \epsilon}[\frac{1}{2\sigma_t^2}||\frac{1}{\sqrt{\alpha_t}}\mathcal{C}(x_t - \frac{\beta_t}{\sqrt{1-\bar{\alpha}_t}}\epsilon, Tp(x_t - \frac{\beta_t}{\sqrt{1-\bar{\alpha}_t}}\epsilon)) - \mu_\theta(x_t, Tp(x_t), x_0)||_2^2]
\end{aligned}
\tag{19}
$$

This suggests that with a given input $x_t$, our diffusion model aims to predict $\frac{1}{\sqrt{\alpha_t}}\mathcal{C}(x_t - \frac{\beta_t}{\sqrt{1-\bar{\alpha}_t}}\epsilon, Tp(x_t - \frac{\beta_t}{\sqrt{1-\bar{\alpha}_t}}\epsilon))$ as following

$$
\begin{aligned}
&\mu_\theta(x_t, Tp(x_t), x_0) \\
&= \mu_\theta(x_t, Tp(x_t), \frac{1}{\sqrt{\bar{\alpha}_t}}(x_t - \sqrt{1-\bar{\alpha}_t}\epsilon_\theta(x_t, Tp(x_t), t)) \\
&= \frac{1}{\sqrt{\alpha_t}}\mathcal{C}(x_t - \frac{\beta_t}{\sqrt{1-\bar{\alpha}_t}}\epsilon_\theta(x_t, Tp(x_t), t), Tp(x_t - \frac{\beta_t}{\sqrt{1-\bar{\alpha}_t}}\epsilon_\theta(x_t, Tp(x_t), t))),
\end{aligned}
\tag{20}
$$

where $\epsilon_\theta$ is learnt by our model aiming at predicting $\epsilon$ of $x_t$. In addition, although the data is involved noise and stochasticity, the topological summaries (e.g., persistence diagrams) are remained stable under perturbation (Kusano et al., 2016; Chazal et al., 2014). In this regard, to sample $\mathcal{C}(x_{t-1}, Tp(x_{t-1})) \sim p_\theta(x_{t-1}|x_t, Tp(x_t))$, we have

$$
\begin{aligned}
\mathcal{C}(x_{t-1}, Tp(x_{t-1})) =& \frac{1}{\sqrt{\alpha_t}}\mathcal{C}(x_t - \frac{\beta_t}{\sqrt{1-\bar{\alpha}_t}}\epsilon_\theta(x_t, Tp(x_t), t), Tp(x_t - \frac{\beta_t}{\sqrt{1-\bar{\alpha}_t}}\epsilon_\theta(x_t, Tp(x_t), t))) + \sigma_t z \\
=& \frac{1}{\sqrt{\alpha_t}}(x_t - \frac{\beta_t}{\sqrt{1-\bar{\alpha}_t}}\epsilon_\theta(x_t, Tp(x_t), t)) + \sigma_t z,
\end{aligned}
\tag{21}
$$

where $z \sim \mathcal{N}(0, 1)$. Using Eq. (21) orderly from $T$ to 0, we can generate synthesis samples from noise. Combining Eq. 20-21, Eq. 19 is simplied as

$$
\mathbb{E}_{q(x_t|x_0)}[D_{KL}(q(x_{t-dt}|x_t, Tp(x_t), x_0)||p(x_{t-dt}|x_t, Tp(x_t))] \leq \mathbb{E}_{x_0, \epsilon}[\frac{\beta_t^2}{2\sigma_t^2\alpha_t(1-\bar{\alpha}_t)}||\epsilon - \epsilon_\theta(x_t, Tp(x_t), t)||_2^2]
\tag{22}
$$

From above, we observe that objective of our Topo-Diffusion model can be transferred into an ELBO. □