# OpenReview forum: "Topo-Diffusion: Topological Diffusion Model for Image and Point Cloud Generation"
_ICLR.cc/2024/Conference — Submitted to ICLR 2024_

### Official Review · Reviewer_yf4E · 2023-10-15

**Soundness:** 4 excellent
**Presentation:** 3 good
**Contribution:** 4 excellent
**Rating:** 8
**Confidence:** 3

**Summary:**

The paper proposes Topo-Diffusion, a diffusion model method for topology while investigating whether topological properties can serve as a reliable signal in the diffusion process. The method learns topological features via layer-wise topological-aware residual blocks. The authors apply persistent homology to reconstruct the underlying geometric and topological structure of data.

**Strengths:**

[+] The paper explores a novel idea of applying diffusion models to topology with topology-specific features.
[+] The authors provide theoretical guarantees of their model.
[+] The authors demonstrate the utility of their approach on real-world data.

**Weaknesses:**

[-] The related works section of diffusion models is missing many key works such as [1,2].
[-] Figure 2/3 should have its axes labeled for greater clarity.
[-] The authors should elaborate on what diverse data types are worth exploring using Topo-Diffusion in their conclusion.


[1] Song, Yang, et al. "Score-based generative modeling through stochastic differential equations." arXiv preprint arXiv:2011.13456 (2020).

[2] Ho, Jonathan, Ajay Jain, and Pieter Abbeel. "Denoising diffusion probabilistic models." Advances in neural information processing systems 33 (2020): 6840-6851.

**Questions:**

This is an overall sound paper, providing a new application for diffusion models.

---

> ### Author Response · Authors · 2023-11-20
> **Response to Reviewer yf4E**
>
> Thank you very much for appreciating the novelty of  the Topo-Diffusion model and constructive suggestions. We have addressed your questions below and also added the respective updates into the revision.
>
> **Response to weakness**:
>
> [-] Thank you very much for your comments! We have included [1] [2] and some other existing works in our paper. We have rewritten the introduction and related work sections with a new perspective, in which we mainly focus on the underlying diffusion mechanisms.
>
> [-] Thank you for your suggestion. We have redrawn Figures 2 and 3.
>
> [-] We appreciate this good suggestion. We summarize that we mainly conducted experiments on structural data (point cloud, cell images, digital images, satellites imagery) to see the effects straightforwardly. We also conducted experiments on real-world human face dataset and the results showed that our proposed Topo-Diffusion is able to enhance real world data too.

---

> > ### Author Response · Authors · 2023-11-22
> > **Additional Response to Reviewer yf4E**
> >
> > Dear Reviewer yf4E,
> >
> > We sincerely appreciate your detailed and constructive feedback of our work, and have carefully considered your comments in our rebuttal response. We would be grateful if you could let us know whether our responses address your concerns. We are eager to engage in any further discussions.

---

> > > ### Comment · Reviewer_yf4E · 2023-11-22
> > > **Thanks for the great responses.**
> > >
> > > No further comments.

---

> > > > ### Author Response · Authors · 2023-11-22
> > > > **Thanks to Reviewer yf4E**
> > > >
> > > > Thank you very much again for your invaluable and constructive feedback!

---

### Official Review · Reviewer_E1SD · 2023-10-27

**Soundness:** 4 excellent
**Presentation:** 3 good
**Contribution:** 4 excellent
**Rating:** 6
**Confidence:** 3

**Summary:**

In this manuscript, a novel diffusion model known as Topo-Diffusion has been proposed. In this model, the topological properties of the data are extracted, and the diffusion process is done within the topological domain. More specifically, by using the persistent homology, both local topological properties and global structural information can be characterized and served as a reliable signal for generating a high-quality sample. The author(s) also provide a theoretical prove of the stability properties of the model. Finally, the mode has been tested on seven real-world and synthetic datasets. It has been found that Topo-Diffusion can outperforms classical diffusion models in various evaluation metrics.

**Strengths:**

This manuscript is to propose a novel diffusion model, known as Topo-Diffusion, which is to employ the classical diffusion model in the special topological domain and sample from reconstructed topological features. The key innovation is to identify data structural features through a topological degrading and reconstruction process and further use them for data generation in the diffusion model.

**Weaknesses:**

Even though TDA is very powerful in characterizing the structural topological information, either local or global, it has limitations in the detailed description of local geometric features. It seems that the current model is more suitable for data with strong topological properties but may fall short when the detailed geometric information also plays an important role.

**Questions:**

1)	The author(s) claim that “we introduce two persistence point transformation methods (see Eq. 2), i.e., (i) birth-lifespan transformation and (ii) differentiable distribution transformation”. These transformation methods “can transform a PD (i.e., a multi-set of persistence points) into a topological feature vector or function which can be easily integrated into any type of models”. However, all the transformation methods used in the paper are just standard Betti curve, persistent landscape, and persistent image. In fact, the so called “birth-lifespan transformation and differentiable distribution transformation” are just rename or summarization of existing methods. In fact, the vectorization of PD is a hot research area and various models have been proposed, including Barcode statistics, Algebraic functions and tropical functions, Binning approach, Persistent codebook, Persistent paths and signature features, 2D/3D representation, etc. Various topological kernels have also been developed. A detailed discussion of these models can be found in various review papers, such as,
Pun, Chi Seng, Kelin Xia, and Si Xian Lee. "Persistent-Homology-based Machine Learning and its Applications--A Survey." arXiv preprint arXiv:1811.00252 (2018).
Dey, T. K., & Wang, Y. (2022). Computational topology for data analysis. Cambridge University Press.
Z. X. Cang, Lin Mu and Guo-Wei Wei, Representability of algebraic topology for biomolecules in machine learning based scoring and virtual screening, PLOS Computational Biology, 14(1), e100592 (2018).
The authors should not overclaim their contributions and properly cite the related works.
2)	The author(s) state that “A notable benefit of the birth-lifespan transformation is its parameter-free definition. This parameter-free Birth-lifespan transformation thus eliminates the necessity for any adjustments, negating the potential for overfitting”.  This is not proper! The transformation can only generate a continuous function, which needs to be further discretized or vectorized. The size of the final topological vector depends on the the process of discretization, and it can be very large if a refined mesh or grid is used!  In contract, if Barcode statistics model or Algebraic functions are employed, the final topological vectors are always of limited sizes. In this way, they are really “parameter-free”.
3)	Figure 19 is very weird. The barcode is not consistent with the data! For instance, in the second simplicial complex (from left to right), it contains 7 0-dim homology generators, but the number of the corresponding 0-dim barcodes is just 6! Further, in the third simplicial complex (from left to right), it also contains 7 0-dim homology generators, but the number of the corresponding 0-dim barcodes is just 1! Moreover, it contains only 2 1-dim homology generators, but the number of the corresponding 1-dim barcodes is 3!
4)	Page 4, “where $K$ denotes the maximum dimension of simplices of $x_t$”. Normally the dimension of PD can be different from the highest order of simplices.
5)	The citations in the paper are very messy.  For instance, Page 1, “Gao et al. Gao et al. propose”; Page 2, “Liu et al. Liu et al. (2019)”…

---

> ### Author Response · Authors · 2023-11-20
> **Response to Reviewer E1SD (1/2)**
>
> Thank you very much for recognizing the novelty of the proposed ideas and the importance of the topic and for the valuable feedback and references! We have addressed your questions below and also added the respective updates into the revision.
>
> **Response to Question 1:** We thank the reviewer for suggesting these papers! We agree with this comment. Here we would like to point out that our proposed Topo-Diffusion is the first approach bridging persistent homology representations of the data and diffusion generative models (i.e., incorporates these two persistence point transformation methods into the diffusion model), and improves the robustness and performance on seven real-world and synthetic datasets. Also, we would like to highlight that our Topo-Diffusion provides flexible architecture which is capable of utilizing various types of topological summaries of the input object. We have cited all the above papers and introduced their methods/applications in the `Related Work’ section.
>
> **Response to Question 2:** Regarding the "parameter-free", we mean that, compared with other deep learning models, it does not have arbitrary hyperparameters that the users have to set in order to perform the analysis. Now we have removed the "parameter-free" and modified this statement as "We introduce explicit constructions of topological summaries for persistence vectorizations through the birth-lifespan transformation using *Betti curve* and *persistence landscape*. Specifically, (i) Betti curve allows us to measure various topological information with a given filtration and does not inherit any implicit biases; and (ii) the advantages of persistence landscape are invertible, stable, and nonlinear." *(see Birth-lifespan transformation in the revision)*.
>
> **Response to Question 3:** Thank you for raising this question. To better illustrate the barcodes based on a sequence of Vietoris-Rips complexes for a point cloud, we have drawn a new figure on a point cloud with 5 points *(please refer to the Figure 19 in the revision)*. As shown in Figure 19, the top three subfigures are snapshots of the evolving complex as threshold $\epsilon$ increases. Note that the positions of the points in the figure do not exactly reflect the interpoint distances. The distances between two points in the point cloud less than or equal to $\epsilon$ are depicted in blue. The $\epsilon$ values corresponding to the two ends of the horizontal bars mark the birth and death times of topological features. We can observe that (i) when $\epsilon=0$, $\beta_0 = 0$ (i.e., $H_0$) and $\beta_1 = 0$ (i.e., $H_1$); (ii) when $\epsilon=0.4$, $\beta_0 = 3$ and $\beta_1 = 0$; and (iii) when $\epsilon=0.8$, $\beta_0 = 1$ and $\beta_1 = 3$.
>
> **Response to Question 4:** We agree with this comment. Following the Reviewer’s suggestion, we have fixed this sentence, i.e., where $K$ denotes the maximum dimension of homological features of $x_t$, and updated it in the revision.
>
> **Response to Question 5:** Thank you so much for notifying us this, we have corrected these issues in the revised version.

---

> > ### Comment · Reviewer_E1SD · 2023-11-22
> > **The response is great**
> >
> > All my comments have been well addressed. I have no further comments!

---

> > > ### Author Response · Authors · 2023-11-22
> > > **Thanks to Reviewer E1SD**
> > >
> > > Thank you very much again for your invaluable and insightful feedback!

---

> ### Author Response · Authors · 2023-11-20
> **Response to Reviewer E1SD (2/2)**
>
> **Response to weakness:** Thank you very much for pointing out this concern. We have conducted additional experiments on three real datasets, which are labeled faces in the wild (LFW) dataset, the circuit reconstruction from electron microscopy images (CREMI) dataset, and NASA satellite image dataset (NASA’s Satellite Imagery) to evaluate our proposed method (which would further strengthen our work and facilitate readers' better understanding of the applications of our Topo-Diffusion). We will conduct and provide more experimental results in the final version. In the revision, we provide results on human face dataset to demonstrate the performance of our Topo-Diffusion to enhance the image quality. Focusing on structure information,  we have also conducted additional  experiments on cell image and satellite image datasets and shown how the structural information in synthetic image can be captured by Topo-Diffusion.
>
> We have listed the following quantitative results to demonstrate the performance of the proposed Topo-Diffusion model. We have provided qualitative results of generated images from CREMI dataset *(see Figure 19 in Appendix)* and the NASA dataset *(see Figure 18 in Appendix)* to demonstrate the effectiveness of including persistence homology information in the Diffusion-based model. The qualitative results show that the persistence homological features enhances the generative model to coverage much richer topological information. The diffusion model can improve the diversity of synthetic data. Topo-Diffusion can improve the quality of generated images in the aspect of capturing topological/geometrical information as well as generating high-quality images.
>
> **Table 1: Performance on the CREMI dataset.**
> |                      |     WGAN   |  Topo-GAN   |   DDPM    |  Topo-Diffusion (ours) |
> |:--------------:|:------------:|:---------------:|:-----------:|:------------------------:|
> |**FID**           |     23.014    |       20.495     |   21.373   |              18.450           |
> | **Precision** |     0.657       |      0.837       |   0.765     |               0.886             |
> | **Recall**      |     0.351       |      0.635       |   0.842     |               0.856             |
> | **RMSE**      |     6.650      |       6.520       |   6.780    |               6.450              |
>
> **Table 2: Performance on the LFW dataset.**
> |                      |     WGAN   |  Topo-GAN   |   DDPM    |  Topo-Diffusion (ours) |
> |:--------------:|:------------:|:---------------:|:-----------:|:------------------------:|
> |**FID**           |     19.820     |           -          |   13.155   |                7.873          |
> |**Precision**  |     0.593       |           -          |   0.595     |               0.864           |
> |**Recall**       |     0.574       |           -          |   0.733     |               0.973           |
> |**RMSE**       |     6.875       |           -          |  4.6011    |               2.974           |
>
> **Table 3: Performance on the NASA’s Satellite Imagery.**
> |                      |     WGAN   |  Topo-GAN   |   DDPM    |  Topo-Diffusion (ours) |
> |:--------------:|:------------:|:---------------:|:-----------:|:------------------------:|
> |**FID**           |     31.020    |           -          |   10.796   |             6.362          |
> |**Precision**     |     0.657       |          -           |   0.789     |             0.869             |
> |**Recall**           |     0.351       |          -          |   0.875     |              0.891             |
> |**RMSE**           |     37.893      |          -         |   13.959   |             7.909             |
>
> We are still training TopoGAN model on LFW dataset, and NASA satellites imagery, and we will update them into the final version once completed.

---

### Official Review · Reviewer_uPRx · 2023-10-30

**Soundness:** 3 good
**Presentation:** 3 good
**Contribution:** 3 good
**Rating:** 6
**Confidence:** 2

**Summary:**

This work introduces Topo-Diffusion, a new denoising diffusion model that efficiently incorporates local topological information.
This works claims the first one that brings the concepts of persistent homology and topological representation learning to diffusion generative models, providing theoretical stability guarantees and improved performance on point clouds and real-world images.

**Strengths:**

The derive theoretical stability guarantees of the proposed topological features of generated data.
Extensive empirical evaluations on both point cloud and read images demonstrate that TopoDiffusion yields state-of-the-art performances.

**Weaknesses:**

Experiments on more complex real world images and point cloud dataset is missing, and current results are limited to toy examples like MNIST and simple point cloud structure.

**Questions:**

refer to the weakness session, I wonder the performance on more complex dataset like shapenet point cloud.

---

> ### Author Response · Authors · 2023-11-20
> **Response to Reviewer uPRx**
>
> Thank you very much for recognizing the novelty of the proposed ideas and the importance of the topic and for the thought provoking question on applying our model to more complex real world datasets! We have addressed your questions below and also added the respective updates into the revision.
>
> **Response to Question**: Thank you so much for pointing this out. We have conducted additional experiments on three real datasets, which are labeled faces in the wild (LFW) dataset, the circuit reconstruction from electron microscopy images (CREMI) dataset, and NASA satellite image dataset (NASA’s Satellite Imagery) to evaluate our proposed method (which would further strengthen our work and facilitate readers' better understanding of the applications of our Topo-Diffusion). We will conduct and provide more experimental results in the final version. In the revision, we provide results on human face dataset to demonstrate the performance of our Topo-Diffusion to enhance the image quality. Focusing on structure information,  we have also conducted additional  experiments on cell image and satellite image datasets and shown how the structural information in synthetic image can be captured by Topo-Diffusion.
>
> We have listed the following quantitative results to demonstrate the performance of the proposed Topo-Diffusion model. We have provided qualitative results of generated images from CREMI dataset *(see Figure 19 in Appendix)* and the NASA dataset *(see Figure 18 in Appendix)* to demonstrate the effectiveness of including persistence homology information in the Diffusion-based model. The qualitative results show that the persistence homological features enhances the generative model to coverage much richer topological information. The diffusion model can improve the diversity of synthetic data. Topo-Diffusion can improve the quality of generated images in the aspect of capturing topological/geometrical information as well as generating high-quality images.
>
> **Table 1: Performance on the CREMI dataset.**
> |                      |     WGAN   |  Topo-GAN   |   DDPM    |  Topo-Diffusion (ours) |
> |:--------------:|:------------:|:---------------:|:-----------:|:------------------------:|
> |**FID**           |     23.014    |       20.495     |   21.373   |              18.450           |
> | **Precision** |     0.657       |      0.837       |   0.765     |               0.886             |
> | **Recall**      |     0.351       |      0.635       |   0.842     |               0.856             |
> | **RMSE**      |     6.650      |       6.520       |   6.780    |               6.450              |
>
> **Table 2: Performance on the LFW dataset.**
> |                      |     WGAN   |  Topo-GAN   |   DDPM    |  Topo-Diffusion (ours) |
> |:--------------:|:------------:|:---------------:|:-----------:|:------------------------:|
> |**FID**           |     19.820     |           -          |   13.155   |                7.873          |
> |**Precision**  |     0.593       |           -          |   0.595     |               0.864           |
> |**Recall**       |     0.574       |           -          |   0.733     |               0.973           |
> |**RMSE**       |     6.875       |           -          |  4.6011    |               2.974           |
>
> **Table 3: Performance on the NASA’s Satellite Imagery.**
> |                      |     WGAN   |  Topo-GAN   |   DDPM    |  Topo-Diffusion (ours) |
> |:--------------:|:------------:|:---------------:|:-----------:|:------------------------:|
> |**FID**           |     31.020    |           -          |   10.796   |             6.362          |
> |**Precision**     |     0.657       |          -           |   0.789     |             0.869             |
> |**Recall**           |     0.351       |          -          |   0.875     |              0.891             |
> |**RMSE**           |     37.893      |          -         |   13.959   |             7.909             |
>
> We are still training TopoGAN model on LFW dataset, and NASA satellites imagery, and we will update them into the final version once completed.

---

> > ### Comment · Reviewer_uPRx · 2023-11-22
> > **Additonal experiments**
> >
> > Thanks for the additional experiments on realworld datasets. I have no further comments!

---

> > ### Comment · Reviewer_uPRx · 2023-11-22
> > **Additonal experiments**
> >
> > Thanks for the additional experiments on realworld datasets. I have no further comments!

---

> > > ### Author Response · Authors · 2023-11-22
> > > **Thanks to Reviewer uPRx**
> > >
> > > Thank you very much again for your invaluable and constructive feedback!

---

### Official Review · Reviewer_jmNy · 2023-11-01

**Soundness:** 3 good
**Presentation:** 2 fair
**Contribution:** 3 good
**Rating:** 6
**Confidence:** 2

**Summary:**

The manuscript integrates topological data analysis (TDA) into diffusion models for point cloud and image generation. The model is called Topo-Diffusion.

**Strengths:**

I could not find any previous work integrating TDA and diffusion models in this manner, so props for creativity. The results uniformly seem improved.

**Weaknesses:**

The paper is not very clearly written and could do with one more round after this one (if it does not make it this time). Suggestions for the rewrite.

1. Persistent homology (PH) etc. are well established in TDA
2. We need to include this information in the forward and back processes in diffusion
3. There are technical challenges that need to be overcome.
4. Forward process: list and qualitatively explain the technical challenges and the approach taken
5. Reverse reconstruction: do the same
6. Explain the payoff

Do this in a pedagogical concept oriented section before the technical section.

**Questions:**

What happened to the ELBO-based approximation at the heart of the diffusion process. Does it get modified when you include PH information. Use https://arxiv.org/abs/2303.00848 to explain if necessary.

---

> ### Author Response · Authors · 2023-11-20
> **Response to Reviewer jmNy**
>
> We thank the reviewers for their constructive and thorough feedback on this Topo-Diffusion manuscript. Below we have addressed each of the comments in detail. We have also included these modifications into the revised manuscript.
>
> **Response to weakness:**
> Thank you for the feedback. As suggested, we have revised the manuscript by including more details and refining the writing of the manuscript. Specifically, we have included more explanations and references of PH/TDA throughout the manuscript (including introduction, related work, preliminaries, and methodology sections). We also would like to highlight our new interpretation of the diffusion process in diffusion models from a perspective of structural information degrading and reconstruction and how those structural information can be represented by PH/TDA., included in the main body of the revised manuscript. In the introduction section, we have summarized the diffusion process from topological degradation and reconstruction. We explained the relationship between the forward process and topological feature degrading. We further apply multiple widely-used TDA features generation methods to analyze persistent homology information in diffusion. In the reverse process, we conduct the same TDA generation approach to reconstruct the image. We have discussed the technique in detail in the experiments section and demonstrated its effectiveness via a straightforward visualization on point clouds and images. We have added multiple quantitative results on various datasets to evaluate the performance of our Topo-Diffusion model.
>
>
> **Response to Question:** We thank the reviewer for bringing this novel aspect to further our understanding of this work! We have transferred the objective of our Topo-Diffusion into a noise-prediction ($\epsilon$-prediction) and analyzed the Topo-Diffusion in the perspective of ELBO. In details, the objective of training our Topo-Diffusion model is optimized towards a noise prediction loss. In this regard, Topo-diffusion is parameterized as a noise-prediction ($\epsilon$-prediction) model: $s_\theta(x_t, Tp(x_t);\lambda)$, where $x_t$ is degraded data in forward process, $Tp(x_t)=[\phi(Dg^{(k)}(x_t))]^K_{k=0}$ is the topological summaries of $x_t$, $\lambda_t$ is noise scheduler. As the topological summaries are remained stable along the diffusion forward process, for our Topo-Diffusion model, the optimization objective can be simplified as $||s_\theta(x_t, Tp(x_t);\lambda)-\nabla logq(x_t|x_0)||\leq\sigma_{\lambda}^{-2}||\hat\epsilon_\theta(x_t,Tp(x_t);\lambda_t)-\epsilon||^2_2$, where $x_0$ is the original data, $\sigma_\lambda$ denotes a parameter associated with noise scheduler. The weighting function of topo-diffusion model is $w(\lambda)\leq\sigma_\lambda^{-2}$, which can be treated as a monotonically increasing function of $t$. Thus, we can qualitatively conclude that by incorporating topological summaries into diffusion mdoel, objective of our Topo-diffusion model equals to a weighted integral of Evidence Lower Bound Objective (ELBO). In addiction, Denoising Diffusion Probabilistic Model (DDPM) is a noise-prediction based model which does not consider any topological information in the degrading and reconstructing process. The objective of DDPM is $\sigma_\lambda^{-2}||\hat\epsilon_\theta(x_t;\lambda_t)-\epsilon||^2_2$, where $w_{DDPM}(\lambda)=\sigma_\lambda^{-2}$. Comparing Topo-Diffusion and DDPM, we have $w(\lambda) \leq w_{DDPM}(\lambda)$. Therefore, incorporating topological information enhances the diffusion process by tightening the weighting function.
> To better understand our Topo-Diffusion model, we have added a section of *``Appendix E: Understanding Topo-Diffusion with ELBO’’* to illustrate our method in the perspective of ELBO. We also have provided mathematical proof to support our demonstration. Please refer to *Appendix E* of our manuscript for more details.

---

> > ### Author Response · Authors · 2023-11-22
> > **Additional Response to Reviewer jmNy**
> >
> > Dear Reviewer jmNy,
> >
> > We sincerely appreciate your detailed and constructive feedback of our work, and have carefully considered your comments in our rebuttal response. We would be grateful if you could let us know whether our responses address your concerns. We are eager to engage in any further discussions.

---

### Meta-Review · Area_Chair_Zyyv · 2023-12-17

**Metareview:**

This paper proposes incorporating persistent homology into a denoising diffusion model. Results on several (toy) datasets are presented. Reviewers agree that, in principle, the approach is novel, but have also identified issues regarding the toy-ish nature of the datasets, as well as the overall presentation of the TDA-related parts of the manuscript. Overall, the initial scores were borderline, and the rebuttal only clarified parts of the issues raised. After reading the paper myself, the AC's assessment is that some of the results are overstated, such as the stability results in Thm. 1. For instance, the statement follows from standard arguments in most stability proofs and does not warrant it's own "Theorem". Also, the writing and exposition in general, of the PH part is in many places very rough and needs improvement (as also noted by reviewers). Considering the reviews and, in particular, the comments raised, the AC's opinion is that the manuscript is (not yet) ready for publication.

**Justification For Why Not Higher Score:**

Reviewers have raised several comments about the toy-ish nature of the experiments and the overall presentation of the work. While some issues have been resolved during the rebuttal, the AC's opinion (after reading the paper) is that many issues remain, and in addition, some results seem heavily overstated. This work clearly has merit, but in its current form, it is not yet ready for publication.

**Justification For Why Not Lower Score:**

N/A

---

### Decision · Program_Chairs · 2024-01-16

Reject